

# Spatial and seasonal variations of aerosols over China from two decades of multi-satellite observations. Part II: AOD time series for 1995-2017 combined from ATSR ADV and MODIS C6.1 for AOD tendencies estimation

Larisa Sogacheva[1*], Edith Rodriguez[1], Pekka Kolmonen[1], Timo H. Virtanen[1], Giulia Saponaro[1], Gerrit de Leeuw[1], Aristeidis K. Georgoulias[2], Georgia Alexandri[2], Konstantinos Kourtidis[2] and Ronald J. van der A[3]

[1]Finnish Meteorological Institute(FMI), Climate Research Programme, Helsinki, Finland
[2]Laboratory of Atmospheric Pollution and Pollution Control Engineering of Atmospheric Pollutants, Department of Environmental Engineering, Democritus University of Thrace, Xanthi, Greece
[3] Royal Netherlands Meteorological Institute (KNMI), De Bilt, Netherlands

* Correspondence to: Larisa Sogacheva (larisa.sogacheva@fmi.fi)

**Abstract**

Understanding long-term trends in aerosol loading is essential for evaluating the health and climate effects of airborne particulates as well as the effectiveness of pollution control policies. Here we introduce a method to construct a combined annual and seasonal AOD long time series using the Along-Track Scanning Radiometers (ATSR: ATSR-2 and AATSR) and MODerate resolution Imaging Spectroradiometer Terra (MODIS/Terra), which together cover the period of 1995-2017. The long-term (1995-2017)
annual and seasonal combined AOD time series are presented for the all of mainland China, for southeastern (SE) China and for 10 selected regions in China and analysed to reveal the AOD tendencies during the last 23 years. Linear regression has been applied to individual L3 (1°x1°) pixels of the annual and seasonal combined AOD time series to estimate the AOD tendencies for three periods: 1995-2006 (P1) and 2011-2017 (P2), as regarding the changes in the emission reduction policies, and the whole period 1995-2017 (WP), when combined AOD time series is available.

Positive tendencies of annual AOD (0.006, or 2% of AOD, per year) prevailed across all of mainland China before 2006 due to emission increases induced by rapid economic development.  In southeastern China, the annual AOD positive tendency in 1995-2006 was 0.014, or 3% of AOD, per year in SE China, reaching maxima (0.020, or 4% of AOD, per year) in Shanghai and the Pearl River Delta regions.

 Negative AOD tendencies (-0.015, or -6% of AOD, per year) were identified across most of China after 2011 in conjunction with
effective emission reduction in anthropogenic primary aerosols, $SO_2$ and NOx (Jin et al., 2016, van der A et al., 2017). The strongest AOD decrease is observed in Chengdu (-0.045, or -8% of AOD, per year) and Zhengzhou (-0.046, or -9% of AOD, per year) areas,



while over the North China plane and coastal areas the AOD decrease was lower than -0.03, or ca -6% of AOD, per year. In the less populated areas, the AOD decrease was small.

The AOD tendencies for the whole period 1995-2017 were much less pronounced compared to P1 and P2. The reason for that is that positive AOD tendency has been observed at the beginning of WP (in P1) and negative AOD tendency has been observed at
the end of WP (in P2), which partly cancel each other during 1995-2017. In the WP, AOD was slightly increasing over the Beijing-Tianjin-Hebei area (0.008, or 1.3% of AOD, per year) and the Pearl River Delta (0.004, or 0.6 % of AOD, per year). A slightly negative AOD tendency (-0.004, or -0.7% per year) was observed in the Chengdu and Zhengzhou areas.

Seasonal patterns in the AOD regional long-term trend are evident. The contribution of seasonal AOD tendencies in annual tendencies was not equal along the year. While the annual AOD tendency was positive in 1995-2006, the AOD tendencies in winter
and spring were slightly negative (ca. -0.002, or -1% of AOD, per year) over the most of China during that period. AOD tendencies were positive in summer (0.008, or 2% of AOD, per year) and autumn (0.006, or 6% of AOD, per year) over all mainland China and SE China (0.020, or 4% of AOD, per year and 0.016, or 4% of AOD, per year in summer and autumn, respectively). The AOD negative tendencies in 2011-2017 were higher compared to other seasons in summer over China (ca. -0.021, or -7% of AOD, per year) and over SE China (ca. -0.048, or -9% of AOD, per year).

The results obtained in the current study show that the effect of the changes in the emission regulations policy in China during 1995-2017 is evident in AOD gradual decrease after 2011. The effect is more visible in the highly populated and industrialized regions in SE China.

## 1 Introduction

Atmospheric aerosols play an important role in climate change through direct and indirect processes. In order to evaluate the effects
of aerosols on climate, it is necessary to study their spatial and temporal distributions. Understanding the long-term changes and the trend in AOD on the Earth, especially in the developing countries like China, becomes increasingly essential for accurately assessing the radiative forcing, as well as better constraining the climate models (Li et al., 2013). The rapid development of industry, the combustion of fossil fuel, the emissions of industrial fumes and contaminated gas lead to a significant increase of atmospheric aerosols, which do not only affect climate, but also constitute a threat to human health (Cao et al., 2017). It is critical for
environmental and epidemiological studies to accurately investigate the fine-scale spatial and temporal changes in aerosol concentrations regarding the industrialization and urbanization (Streets et al., 2009).

Satellite aerosol remote sensing is a rapidly developing technology that may provide good temporal sampling and superior spatial coverage to study aerosols. The most common parameter derived from satellite observations is the Aerosol Optical Depth (AOD), which is a measure of the extinction of electromagnetic radiation at a given wavelength due to the presence of aerosols integrated
over the atmospheric column. AOD is a key factor for the estimation of the aerosol concentration, the evaluation of atmospheric conditions and the effect of atmospheric aerosols on climate.



The air pollution in China is severe (Bouarar et al., 2017), widely distributed, and the atmospheric chemical reactions complex (Kulmala et al., 2015). The strong economic growth in China (http://www.worldbank.org/en/country/china/overview) has significantly raised the living standards (http://www.china.org.cn/opinion/2013-10/24/content_30391004.htm), but it has also brought serious environmental damage and degradation (Tang et al., 2015).

In 1970–1990, the dominant contributing sources were big and small coal burning stoves widely used in power plants, industry, utilities and households (Jin et al., 2016). Coal smoke mainly contains sulphuric acid ($SO_2$), total suspended particle matter (TSP), but also nitrogen oxides ($NO_X$) and carbon monoxide (CO). Other sources, such as dust from construction sites, mainly consisting of primary PMs, contributed less. During the transition period 1990–2000, the growing number of vehicles, mainly in megacities contributed a lot to the increase in $NO_x$ and volatile organic compounds (VOCs). From 2000 to the present, the anthropogenic air
pollution is ingrained in the megacities and spreading to the regional level (Jin et al., 2016). However, national regional plans, environmental laws, rules and standards exist in China, which are revised with the Five-Years-Plans, which are series of social and economic development initiatives (Jin et al., 2016).

The further urbanization is a consequence of the growing industry. Over the period 1978–2016, more than 550 million migrants moved to China's cities, resulting in a large rise of urban population from 18 to 57% (Zhang, 2017). Coastal regions where
manufacturing and services are better developed, especially big cities, are the major destinations (Chan, 2012). During the last two decades, a strong population inflow to eastern China has been reported (Ma and Chen, 2012; Center for International Earth Science Information Network - CIESIN - Columbia University, 2017). The population in the North China Plane, the Yangtze River Delta and the Pearl River Delta was steadily increasing mainly due the growth of large cities, or metropolises (Fig. 1). The population density has increased In Beijing-Tianjin-Hebei area, Shanghai, Xiamen and Guangzhou, by more than 200% during 2000-2015,
while in Wuhan, Chengdu and Zhengzhou the population has grown nearly by 50%. Such a strong population growth has resulted in the fast urbanization and the further industrialisation and infrastructure development in those regions.




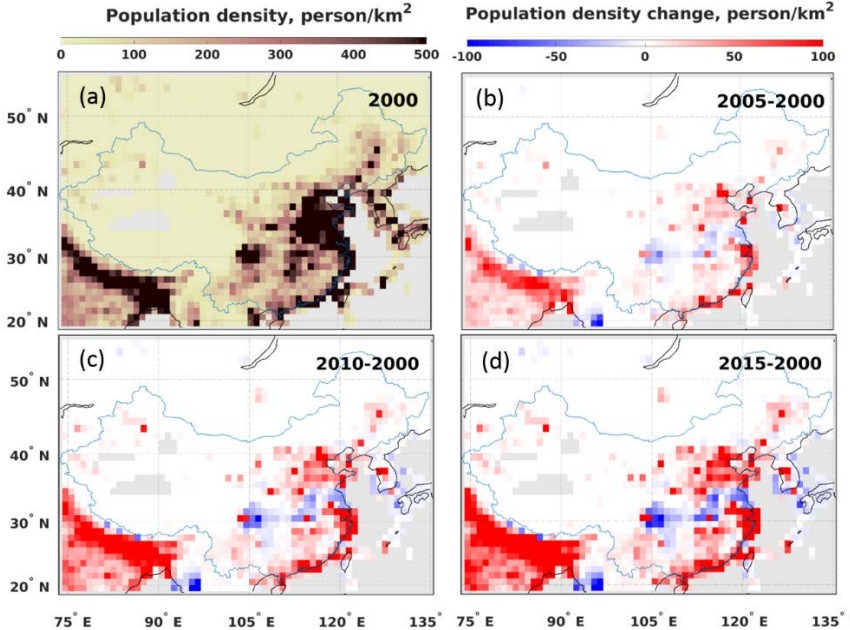

**Figure 1. The population density in China in 2000 (a) and the population density change in 2005 (b), 2010 (c), 2015 (d) comparing with 2000. Data are obtained from https://doi.org/10.7927/H4DZ068D, accessed 09.02.2018.**

Due to the sparse surface network in China and limited data availability, the increasing attention has been paid over the last decades to the satellite aerosol remote sensing to study the long-time changes in aerosol properties over China. Xie and Xia (2009) reported statistically increasing AOD trends in spring and summer in north China annually during the period from 1982 to 2001 using the Total Ozone Mapping Spectrometer (TOMS). Su et al. (2010) analyzed the AOD distribution over 10 locations in East Asia using the yearly mean POLDER-AOD products during the period from 2005 to 2009. Guo et al. (2012) analyzed the monthly AOD trends

of TOMS and MODIS in China during the period from 1982 to 2006. The temporal and spatial trends in AODs were analyzed with MODIS data (2000-2008) over eight typical regions in China (Guo et al. 2011). Opposite in sign AOD trends in different locations in China have been recorded using observations from the Aerosol Robotic Network (AERONET, Holben et al., 1998) by Li et al. (2014). Many authors reported decreasing trends in the United States and Europe, and increasing trends in China in the last few years, as summarized in Zhao et al. (2017), while a few recent studies (Zhang et al., 2016; He et al., 2016; Mehta et al., 2016)

showed decreasing trends in China over the last decade.

The expected lifetime of the satellites is about 10-15 years. To study the longer trends of the substances using satellites, the information from different satellites should be combined. Combining multiple sensors could increase the period for data availability and reduce data uncertainties (Li et al., 2016).

In this study, we introduce the method to combine the Along-Track Scanning Radiometers (ATSR-2 and AATSR, hereunder referred

as ATSR) and MODerate resolution Imaging Spectroradiometer Terra (MODIS/Terra) seasonal and annual aerosol data for the



period 1995–2017 to investigate AOD tendencies in China during more than two decades. The method is based on the results from the comparison between ATSR and MODIS AOD products (Sogacheva et al., 2018, further referred as Part I). We also investigate whether the tendencies in AOD are related to emission changes as well as to the pollution control policies in China. Since spatial difference exists in aerosol loading in China (Part I, de Leeuw et al., 2018), we present the combined AOD long time series and

estimate AOD tendencies for different regions in China.

The objectives of this study are: (1) to combine AATSR ADV and MODIS seasonal and annual aerosol data during 1995–2017 and (2) to analyze the spatial and temporal seasonal and annual aerosol long-time series and link the AOD tendencies to emission control policies in China.

The paper is structured as follows. The AATSR and MODIS/Terra AOD products are introduced in Section 2, including the short

description of the instruments, aerosol retrieval algorithms, data sets and validation results. The study area is discussed in Sect. 3, where 10 selected regions are introduced. In Sect 4, a method used to construct the combined ATSR (1995-2011) and MODIS (2000-2017) AOD time series for more than 20 years (1995-2017) is presented, AOD correction is discussed, long-term annual and seasonal combined AOD time series are shown.  In Sect. 5, the changes in the emission reduction policies in China are discussed; the AOD tendencies are estimated for three periods related to the changes in the emission policies in China and discussed. The main

conclusions are summarised in Sect.6.

## 2. AOD data: instrumentation, AOD retrieval algorithms and aerosol data sets

Below we briefly introduce ATSR and MODIS instruments, AOD retrieval algorithms and aerosol data sets. For more details, see de Leeuw et al. (2018) and Part I.

### 2.1 ATSR ADV

The ATSR instruments, ATSR-2 on board the European Remote Sensing satellite ERS-2 (1995-2003) and AATSR on board the environmental satellite Envisat (2002-2012), referred hereunder as ATSR, were developed to provide the high-accuracy measurements of the Sea Surface Temperature (SST) and are also successfully used for atmospheric aerosol retrievals (e.g., Flowerdew and Haigh, 1996, Veefkind et al., 1998, Sayer, 2010, Kolmonen et al., 2016). Together, these instruments provided 17 years of global data. Both satellites flew in a sun-synchronous descending orbit with a daytime equator crossing time of 10:30 LT

(ERS-2) and 10:00 LT (ENVISAT). The ATSR is a dual view instrument. One view is near-nadir and the other one is at a 55º forward angle.  The time between the two views is approx. 150 seconds along track. The nominal resolution at nadir is 1x1 km$^2$ and the swath width is 512 km, which results in global coverage in 5-6 days. ATSR has three wave bands in the visible – near infrared (centred near 555 nm, 659 nm, 865 nm) and four bands in the mid- to thermal infrared (centred near 1600 nm, 3700 nm, 10850 nm, 12000 nm).



Over land, the AATSR Dual View (ADV) AOD retrieval algorithm uses the two ATSR views simultaneously to eliminate the contribution of land surface reflectance to the TOA radiation to retain the path radiance in cloud-free scenes (Veefkind et al., 1998, Kolmonen et al., 2016, Sogacheva et al., 2017) following Flowerdew and Haigh (1996).

ATSR-2 AOD data are available for the period June 1995- December 2003, with some gaps in 1995 (from January to May, globally) and 1996 (from January to June, globally), while also toward the end (approx. from autumn 2002) the data are less reliable. AATSR data are available for the period May 2002 - April 2012, but some data are missing in 2002 and therefore we use AATSR data only from August 2002 forward. The consistency between the ATSR-2 and AATSR data sets has been discussed in Popp et al. (2016). The ATSR temporal and spatial coverage, as regarding seasonal and annual AOD aggregates over China, is discussed in details in Part I.

The L3 (averaged on a grid of 1º×1º) seasonal aggregates were obtained for winter (DJF), spring (MAM), summer (JJA) and autumn (SON) by averaging the monthly aggregates to corresponding seasons. The annual AOD data were obtained by averaging the monthly AOD data. Hereunder, the ATSR ADV AOD product will be referred to as ADV.

## 2.2 MODIS

The MODIS/Terra (MODerate resolution Imaging Spectroradiometer) sensor (Salomonson et al., 1989) aboard NASA's Terra satellite has been flying in a near-polar sun-synchronous circular orbit for more than fifteen years since December 1999. MODIS/Terra has a daytime equator crossing time at 10:30 LT (descending orbit), a viewing swath of 2330 km (cross track) and provides near-global coverage on a daily basis. Its detectors measure 36 spectral bands between 0.405 and 14.385 µm, and it acquires data at three spatial resolutions (250m, 500m, and 1000m).

MODIS AOD is retrieved using two separate algorithms, Dark Target (DT) and Deep Blue (DB). In fact, two different DT algorithms are utilized, one for retrieval over land (Kaufman et al., 1997; Remer et al., 2005; Levy et al., 2013) and one for retrieval over water surfaces (Tanré et al., 1997; Remer et al., 2005, Levy et al., 2013). The DB algorithm (Hsu et al., 2004, 2013) was traditionally used over bright surfaces where DT performance is limited (e.g. deserts, arid and semi-arid areas) and was further developed for returning aerosol measurements over all land types (Sayer et al., 2014).

In this paper, MODIS/Terra AOD C6.1 DTDB merged (L3) data (2000-2017, https://ladsweb.modaps.eosdis.nasa.gov/ ) are used, which are slightly different from C6 over snow and sea ice, where the cloud mask has been improved (https://earthdata.nasa.gov/earth-observation-data/near-real-time/download-nrt-data/modis-nrt#ed-collection-61).    Such improvement is important over northern and central elevated China, which is periodically or all year around covered by snow. Hereafter, the MODIS/Terra AOD C6.1 DTDB merged AOD product will be referred to as MODIS.

## 2.3 ADV and MODIS validation results.

AOD validation results over China for ADV (de Leeuw et al., 2018) and MODIS C6.1 (Part I) are briefly summarized below.

To evaluate the quality of the ADV and MODIS AOD, these products were compared with reference ground-based AOD data available from AERONET sites (Holben et al., 1998) in the study area (see Table 1 in de Leeuw et al. (2018) for AERONET




locations used for validation). For this comparison, collocated data, i.e. satellite data within a circle with a radius of 0.125° around the AERONET site, were averaged and compared with averaged AERONET data measured within ± 1 hour of the satellite overpass time (Virtanen et al., 2018). The ADV and MODIS AOD validation results are summarised in Table 1.

In this validation exercise, all available collocations between ADV and AERONET and between MODIS and AERONET were

considered ("All points"). The results show that the MODIS algorithm performs slightly better than ADV. The correlation coefficients are 0.88 for ADV and 0.92 for MODIS. Both algorithms show similar (in absolute numbers) bias, which is negative for ADV and positive for MODIS. AOD standard deviation (δ) and root-mean-square (rms) are slightly lower for MODIS. Note, that since MODIS has better coverage as compared to ADV (Part I), the number of validation points for MODIS is considerably larger (4963 and 1132 for MODIS and ADV, respectively).

**Table 1. ADV and MODIS AOD validation results over China for the overlapping period 2000-2011. Statistics (number of points (N), correlation coefficient (R), bias, standard deviation (σ) and root-mean-square (rms)) for all validation points and collocated points (\*when ADV and MODIS overpasses are close in time and ADV, MODIS and AERONET retrieve AOD) are shown. For collocated points, statistics are aggregated also seasonally for winter (DJF), spring (MAM), summer (JJA)**
**and autumn (SON).**

|  | N | | R | | bias | | σ | | rms | |
|---|---|---|---|---|---|---|---|---|---|---|
|  | ADV | MOD | ADV | MOD | ADV | MOD | ADV | MOD | ADV | MOD |
| **All points** | | | | | | | | | | |
|  | 1132 | 4963 | 0.88 | 0.92 | -0.07 | 0.06 | 0.007 | 0.003 | 0.24 | 0.20 |
| **Collocated\* Points:** | | | | | | | | | | |
| **All collocated points** | 255 | | 0.92 | 0.93 | -0.11 | 0.06 | 0.01 | 0.008 | 0.17 | 0.16 |
| **Collocated points, seasons:** | | | | | | | | | | |
| **DJF** | 10 | | 0.92 | 0.96 | -0.04 | -0.17 | 0.023 | 0.052 | 0.10 | 0.19 |
| **MAM** | 87 | | 0.81 | 0.81 | 0.00 | 0.13 | 0.012 | 0.013 | 0.16 | 0.14 |
| **JJA** | 73 | | 0.94 | 0.96 | -0.13 | 0.13 | 0.029 | 0.017 | 0.25 | 0.22 |
| **SON** | 85 | | 0.92 | 0.88 | -0.02 | 0.05 | 0.007 | 0.009 | 0.10 | 0.09 |

To compare the performance of two algorithms when both ADV and MODIS retrieve AOD, we carried out the AOD validation for the cases, where difference between ATSR and MODIS/Terra overpasses was less than 90 minutes and both ADV and MODIS have

successfully retrieved AOD around AERONET ("collocated points"). Altogether, 255 collocations exist between ADV, MODIS and AERONET for the ATSR and MODIS/Terra overlapping period (2000-2011) over China. Validation was done for all collocated points and respectively for four seasons: DJF, MAM, JJA, and SON. For all collocated points, the correlation coefficients (R) were similar for ADV and MODIS (0.92 and 0.93, respectively), ADV was biased negatively (-0.11), while MODIS was biased positively (0.06). In winter, MODIS show a strong negative bias in AOD (-0.17), while correlation for MODIS is higher than for ADV (0.96

and 0.92, respectively). Note, that the number of collocated points is low (10). In spring, R was the same (0.81) for MODIS and





ADV, while bias is 0 for ADV and 0.013 for MODIS. Interestingly, both MODIS and ADV show similar tendency of underestimation of AERONET AOD for AOD >0.6 in spring. In summer, R was a bit higher for MODIS; bias was equal for ADV and MODIS in absolute numbers but opposite in sign (negative for ADV and positive for MODIS). In autumn, R was a higher for ADV (0.92 against 0.88 for MODIS), bias was negative for ADV (-0.02) and positive for MODIS (0.05).

To conclude, MODIS shows better performance for "all points" selection. Similar (in absolute numbers) bias, which is negative for ADV (-0.07) and positive for MODIS (0.06) will be considered in Sect. 4, where method for combining two datasets will be presented. For collocated points, the better performance of either ADV or MODIS is not clearly pronounced.

## 3. Study area and selection of the regions

China covers a huge territory with significant regional differences (Wang et al., 2017), with corresponding differences among

regions by means of land urbanization (Lin et al., 2015). The contradictions between economic growth and environmental quality have varying dimensions in different regions. More urbanized areas have faster rates of growth and are probably the major contributors to air pollution. The highest population density is observed in the northeast, followed by the east (Fig.1). Cities with rapid land urbanization are mainly distributed in the coastal regions and scattered throughout the inland regions. The largest agricultural output provinces are Henan, Sichuan, Hunan, Anhui, and Jiangsu with low population growth and Shandong,

Heilongjiang, and Hubei with balanced urbanization and agricultural development. Sparsely populated areas are provinces in the western China. Thus, from a regional perspective, there are large differences in the levels of economic development in China and the efficiency of the Eastern region is far higher than that of the Central and the Western regions. Additionally, the gap between them tends to expand (Yang and Wang, 2013).

Thus, to study the spatio-temporal tendency of  AOD in China during the last more than 20 years and to examine if the differences

in economic activity among the regions  are reflected in AOD level and tendencies, we focused on the entire area of mainland China, SE China and ten  typical areas, as shown on Fig. 2 (adapted from Part I). The mainland China is the area within the Chinese border indicated by the blue line. SE China, defined in this study as the over-land area between 20º - 41º N and 103º – 135º E, is indicated by purple lines. Numbers indicate the ten study regions. Regions 1-7 nearly cover the SE China.  Region 8 covers the Taklamakan desert, region 9 is over the Tibetan Plateau, and region 10 is over the NE China. Note that all areas used in this study only consider

the AOD over mainland China, i.e. AOD over the oceans or islands is not included.





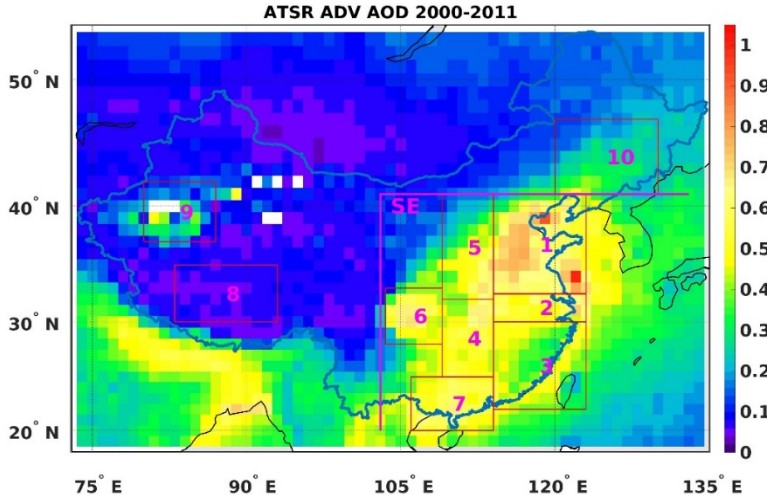

**Figure 2. Regions over mainland China selected for further study of seasonal, interannual and long term behaviour of the AOD, overlaid on the ATSR-retrieved (ADV version 2.31) 12-year aggregated AOD map. Mainland China is indicated with the blue line. The figure shows 10 regions over China and a larger area over SE China indicated with SE (as in Part I).**

With some deviations, the choice of the regions is similar to those in other studies (e.g., Luo et al., 2014, Wang et al., 2017). This choice covers major urban/industrial regions such as the Beijing-Tianjin-Hebei (BTH), the Yangtze River Delta (YRD) and the Pearl River Delta (PRD), Sichuan/Chongqing as well as cleaner regions in the north (region 10 in Fig. 2) and southeast (region 3). In addition, the Tibetan Plateau and Taklamakan desert regions were chosen to represent the sparsely populated and less developed, in terms of industrialization.

## 4. Long-term (1995-2017) annual and seasonal AOD time series combined from ADV and MODIS

Here we introduce a method to combine the AOD data from the ATSR (1995-2011) and MODIS/Terra (2000-2017) radiometers, which together cover the period 1995-2017. The results of ADV and MODIS AOD datasets comparison in Part I are used here to construct a combined AOD dataset. Below, some conclusions obtained in Part I are highlighted.

(1) Similar AOD patterns are observed by ADV and MODIS in yearly and seasonal aggregates (Part I). However, the ADV-MODIS difference maps (Part I, Fig. 7 (right) and Fig. 10 (right)) show that MODIS AOD is generally higher than that from ADV.

(2) The time series in Figs. 9 and S1 (Part I) show large differences between regions, for both sensors, while the interannual patterns in the time series are similar for both ADV and MODIS.

(3) Similar patterns exist in year-to-year ADV and MODIS annual AOD tendencies in the overlapping period (Part I, Tabs. 2 and S2).





(4) ADV and MODIS validation with AERONET (Part I, Sect.3.3) show similar high correlation (0.88 and 0.92, respectively), while the bias is of similar magnitude but opposite in sign: positive for MODIS (0.06) and negative for ADV (-0.07).

**4.1 Method**

The combined time series AOD$_{comb, 1995\text{-}2017}$ have been compiled from AOD estimated for three periods: the first period (T1) is the pre-EOS period (1995-1999), when only ATSR was available, the second period (T2) is the ATSR and MODIS/Terra overlapping period (2000-2011) and the third (T3) is the post-ENVISAT (2012-2017) period, when only MODIS/Terra was available:

$$AOD_{comb,1995-2017} = [AOD_{T1}, \ AOD_{T2}, \ AOD_{T3}] \ . \ (1)$$

First, we introduce the combined AOD for the overlapping period T2 ($AOD_{T2,year}$), when AOD for both ADV and MODIS is available. AOD for each year ($AOD_{T2,year}$) is calculated as a mean AOD of ADV and MODIS:

$$AOD_{T2,year} = \frac{AOD_{MOD,year} + AOD_{ADV,year}}{2} \ . \ (2)$$

The simple averaging has been applied, since ADV and MODIS show similar biases of different signs.

Using ADV and MODIS yearly AOD from the T2, the AOD correction ($AOD_{corr}$) is calculated as the mean difference between ADV and MODIS for the overlapping period:

$$AOD_{corr} = \frac{\sum_{years} \frac{AOD_{MOD,year} - AOD_{ADV,year}}{2}}{N_{years}} \quad . \quad (3)$$

For ADV and MODIS respectively, the AOD correction has been scaled by corresponding AOD, averaged over the overlapping period:

$$AOD_{rel\_corr,ADV} = {AOD_{corr}} \Big/ {mean(AOD_{ADV,2000\text{-}2011})}, \qquad (4)$$

$$AOD_{rel\_corr,MODIS} = {AOD_{corr}} \Big/ {mean(AOD_{MODIS,2000\text{-}2011})} \ . \qquad (5)$$

For T1 and T3, the AOD relative correction ($AOD_{rel\_corr}$) has been applied as positive correction for usually lower ADV AOD (eq. 6) and negative correction for usually higher MODIS AOD (eq. 7)

$$AOD_{T1,year} = \ AOD_{ADV,year} * (1 + AOD_{rel\_corr,ADV}) \ , \ (6)$$

$$AOD_{T3,year} = \ AOD_{MOD,year} * (1 - AOD_{rel\_corr,MODIS}) \ (7).$$

A similar type of merging of data from two separate instruments has been used e.g., Bourassa et al. (2014) to create a long ozone time series.



The method introduced above has been applied pixel-wise to annual and seasonal AOD aggregates from ADV and MODIS to construct the combined AOD time series. For each season, $AOD_{rel\_corr}$ has been computed separately, as introduced for yearly AOD.

**4.2 AOD relative correction**

The spatial distribution of the relative correction for MODIS AOD ( $AOD_{rel\_corr,MODIS}$ ) is shown in Fig.3 (for annual AOD aggregates) and in Fig.4 (for seasonal AOD aggregates). As expected, the highest AOD correction (30-40%) corresponds to the areas, where the agreement is lower between ADV and MODIS AOD ( Figs. 7 and 10 in Part I), e.g., over the bright surface areas, such as Tibetan Plateau and the Taklamakan and Gobi Deserts and Harbin area. The reasons for disagreement between ADV AOD and MODIS AOD is discussed in detail in Part I. Over SE China, the AOD correction is lower (between 10% and 20%). ADV AOD

correction shows similar spatial patterns and thus is not discussed here.

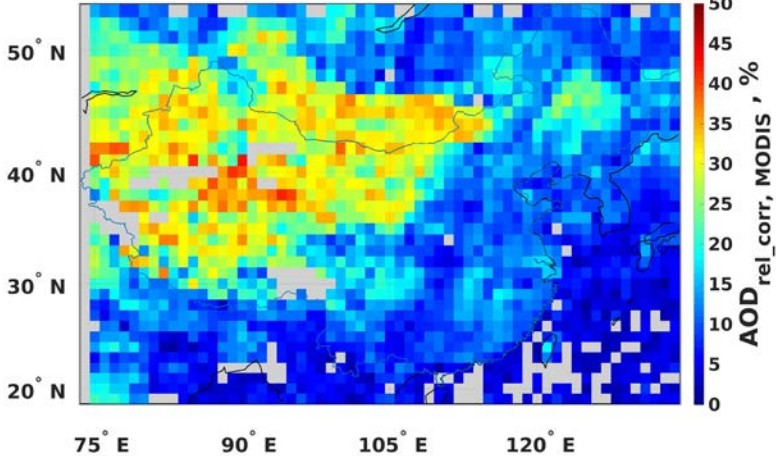

**Figure 3. AOD relative correction for MODIS annual AOD for the combined AOD dataset obtained with the method introduced in Sect.4.1**

Similar to annual, the seasonal AOD correction spatial patterns are recognized. In summer, the correction is smaller, since in that season the agreement in ADV and MODIS AOD is better compared to other seasons (Figs. 7 and 10 in Part I). For seasonal aggregates, the highest correction (ca 45%) is obtained over the Tibetan Plateau in autumn, when AOD is lower compared to spring and summer (Fig. 10 in Part I).

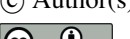



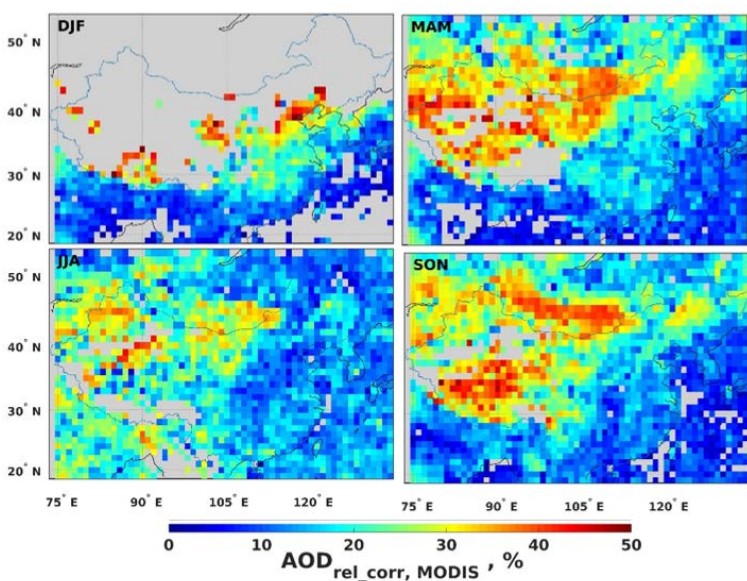

**Figure 4. AOD relative correction for MODIS seasonal AOD for the combined AOD dataset obtained with the method introduced in Sect.4.1**

**4.3 Long-term (1995-2017) yearly and seasonal AOD time series combined from ADV and MODIS**

5    In this section, we introduce the ADV, MODIS and resulted combined annual and seasonal AOD time series for China and SE China. The long-term time series of the annually averaged AOD from ADV (red circles), MODIS (green circles) and combined (yellow rhombs) from ADV and MODIS are shown for the mainland China and SE part of all mainland China in Fig.5. Since ADV was negatively biased and MODIS is positively biased with respect to AERONET AOD, and the biases are similar in absolute value, the combined time series in pre-EOS and post-Envisat periods are practically corrected by increasing ADV and lowering MODIS

10   AOD with the AOD correction as introduced in Sect. 4.1. As expected, the similar interannual variations in the separate datasets are reproduced in the combined time series.

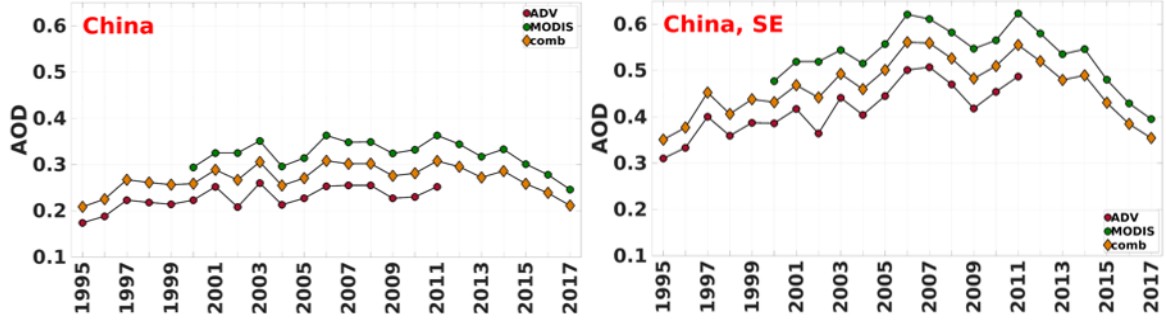

**Figure 5. ADV (red circles), MODIS (green circles) and combined from ADV and MODIS (yellow rhombs) annual AOD time series for China (left) and SE China (right). The method used to combine the ADV and MODIS time series in presented in Sect. 4.1.**



An increase in AOD is observed in China from 1995 towards 2006, with a relative peak in 2003, when active forest fires, which occurred over Russia, strongly increased the AOD in NE China. Between 2006 and 2011, the AOD was not showing a clear tendency. After 2011, the AOD started to steadily decrease, with the exception for year 2014. In SE China, the AOD tendencies are more pronounced and show similar temporal behavior.

In Fig. 6, the long-term time series of the averaged over the seasons (winter, DJF; spring, MAM; summer, JJA; autumn, SON) AOD from ADV (red circles), MODIS (green circles) and combined (yellow rhombs) from ADV and MODIS are shown for all mainland China and SE part of the mainland China. For all seasons, except spring, the difference between ADV and MODIS AOD was low, thus the combined time series closely reproduce both ADV and MODIS AOD. In spring, the combined time series show, on average, 0.1 lower AOD as compared to MODIS (which, in relative numbers, is about 10-20% lower than MODIS AOD). However, as for annual aggregates, for seasonal combined AOD the interannual variability in the time series is similar for both ADV and MODIS. The annual and seasonal AOD tendencies for the combined time series are estimated in Sect. 5.



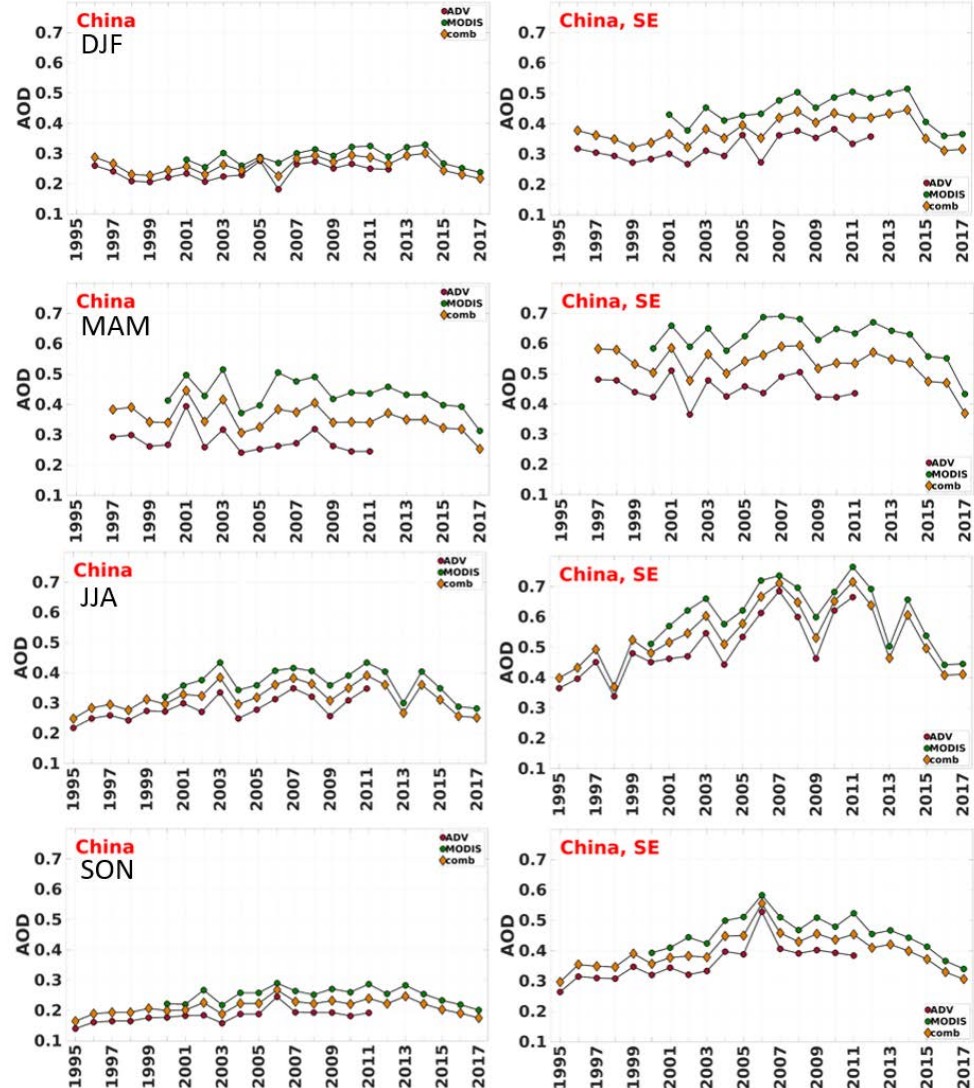

**Figure 6.** ADV (red circles), MODIS (green circles) and combined from ADV and MODIS (yellow rhombs) seasonal AOD time series for China (left) and SE China for winter (DJF), spring (MMA), summer (JJA) and autumn (SON). The method used to combine the ADV and MODIS time series in presented in Sect. 4.1.

## 5. AOD tendencies during 1995-2017

As seen from the combined AOD time series in Figs. 5 and 6, the annually and seasonally averaged AOD in China was increasing from the first year of the AOD available in our analysis (1995) until 2006 (with some delay for different regions). Between 2006 and 2011, AOD was slightly fluctuating. After 2011, the AOD started to decrease until the end of the study period in 2017. To find



out if AOD tendencies relate to the emissions regulation in China during that period, we present a short overview of the emission regulation policies during the last two decades.

## 5.1 Emissions regulation policies in China during the last two decades.

Jin et al. (2016) showed that in China: (1) the early policies, until 2005, were ineffective at reducing emissions; (2) during 2006–
2012, new instruments, which interact with political incentives, were introduced in the 11th Five-Year Plan. However, emission control policies on air pollution have not been strongly reconsidered in the 11th Five-Year Plan, thus no significant reduction of the air pollution has been observed. Regional air pollution problems dominated by fine particulate matter ($PM_{2.5}$) and ground level ozone ($O_3$) emerged. Nevertheless, the national goal of reducing total sulphur dioxide ($SO_2$) emissions by 10% was achieved.

   Jin et al. (2016) also showed that $SO_2$ emissions, as well as smoke and dust emissions, have been gradually decreasing since 2006.
However, the reduction of the total emission of $SO_2$, a single primary pollutant, does not necessarily improve air quality. NOx emissions continued to grow which is explained by the growing number of vehicles mainly in megacities. Total NOx emissions in East China reached their peak levels in 2012, and have stopped increasing since then (van der A et al., 2017), when filtering systems were installed, mainly at power plants but also for heavy industry. These regulations for NOx were announced in 2013 in the Air Pollution Prevention and Control Action Plan (CAAC, 2013).

The "total control" of $SO_2$ and $NO_X$ is strengthened and accelerated in China (Zhang et al., 2017). As an example, the change in standards for cars in the period 2011–2015 reduces the maximum allowed amount of NOx emissions for on-road vehicles by 50 % (Wu et al., 2017). Further emission control scenarios exist in China to control the entirely coal-burning thermal power plants exists. The newly designed control policies considered in Wang et al. (2018) are predicted to lead to reductions in January levels in Beijing between −8.6% and −14.8% for $PM_{2.5}$, $PM_{10}$, $NO_2$, and $SO_2$. However, regional differences in emission control exist. More strict
regulations for on-road vehicles (e.g. a ban on older polluting cars) were introduced on a city level, e.g. in Beijing, rather than nationwide (van der A et al., 2017).

In June 2013, the State Council issued the Action Plan for Air Pollution Prevention and Control. This document laid out the roadmap for air pollution prevention and control across China for 2013-2017. In 2016, the second report was published (http://cleanairasia.org/wp-content/uploads/2016/08/China-Air-2016-Report-Full.pdf) showing the considerable improvements in
emission reduction and air quality. The air quality mostly improved at the developed regions. This report finds that the cities that failed to attain the 2015 air quality target and show slow progress and still suffer from poor air quality are concentrated in Henan Province, Shandong Province, and Northeast China. Compared with the more developed regions, these cities had less experience and insufficient capacity in air pollution prevention and control. Such regional differences might result in some deviation of regional emission tendencies compared to those averaged over the whole China.

Thus, three periods, closely related to the Five-Year-Plans (before 2006, 2006-2011, and after 2011), can be identified, when the emission reduction policies in China are reconsidered.



## 5.2 AOD tendencies for the selected periods: 1995-2006, 2011-2017 and 1995-2017.

Linear regression has been applied to individual L3 pixels of the combined AOD time series to estimate the AOD tendencies over China for three periods: 1995-2006 (P1) and 2011-2017 (P2), as regarding the changes in the emission reduction policies, and the whole period 1995-2017 (WP), when the combined AOD time series is available. The AOD tendencies (dAOD) averaged over the

5   years have been estimated, since P2 is too short to estimate the decadal tendencies. The statistical significance for the tendencies has been estimated with the t-test (Chandler and Scott, 2011). The results were considered significant at p-value<0.05. We also estimated relative tendencies, which are the ratio of tendencies to the corresponded time series averages (Schönwiese and Rapp, 1997).

### 5.2.1 Annual AOD tendencies over China.

10   The AOD tendencies and AOD relative tendencies for annual aggregates for P1, P2 and WP are shown in Figs.7 and 8, respectively. AOD positive tendencies are coloured with red, AOD negative tendencies are coloured with blue. The pixels are marked in Fig. 7 with the green dots when the linear fit was giving the statistically significant results (p-value<0.05).

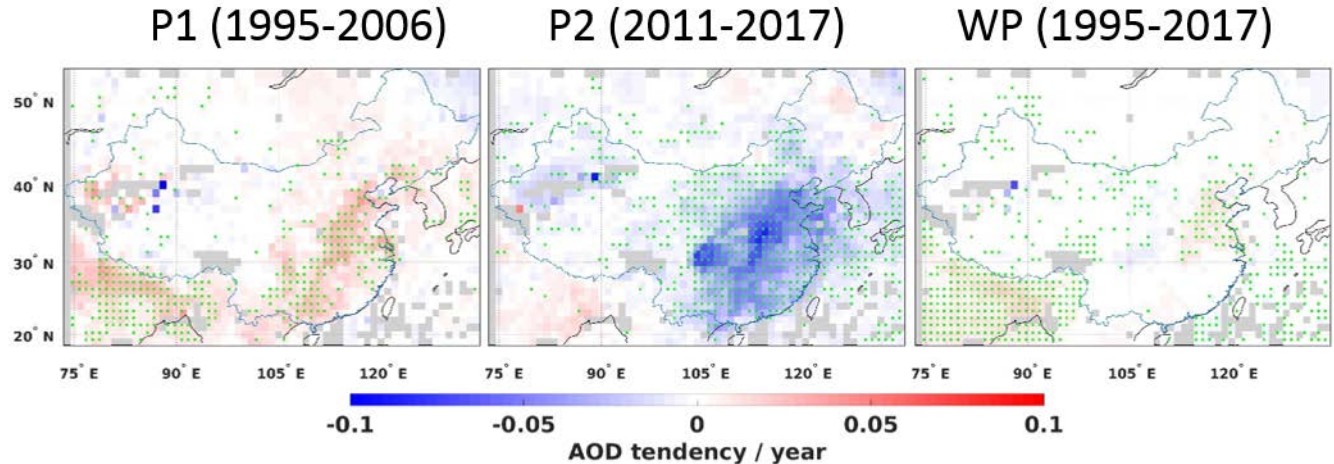

**Figure 7. AOD tendencies (per year, see colorbar) from annually aggregated combined AOD time series for three periods: 1995-2006 (P1),**
15  **2011-2017 (P2) and 1995-2017 (WP). Individual pixels are marked with the green dots when the linear fit was giving the statistically significant results (p-value<0.05).**





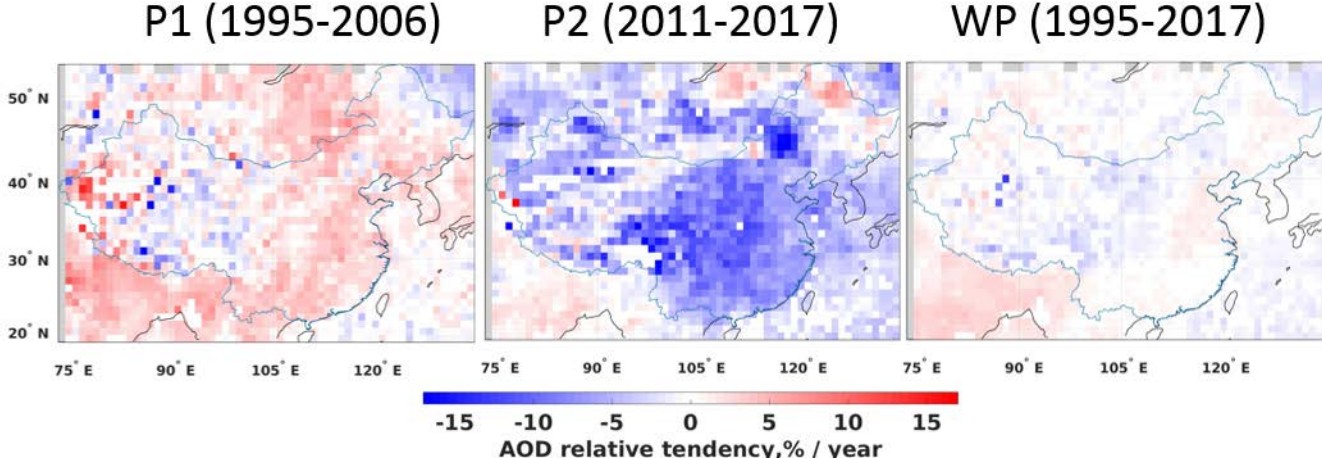

**Figure 8. AOD relative tendencies (per year, see colorbar) from annually aggregated combined AOD time series for three periods: 1995-2006 (P1), 2011-2017 (P2) and 1995-2017 (WP).**

A statistically significant increase in AOD of 0.02 to 0.04 per year (Fig.7, left panel), which is about 3-6% of AOD per year (Fig.8, left panel), was observed in P1 over SE China. Similar increase of AOD was recognized over the Taklamakan Desert. However, since the annual AOD over the Taklamakan Desert is lower compared to the SE China (Fig. 7 in Part I), the relative tendency over the Taklamakan Desert is higher (up to 10% per year for certain pixels) than over the SE China for annual AOD. Note that for only few pixels over the Taklamakan Desert the linear fit provides the statistically significant results. Over western and northern China,

the AOD tendencies were slightly negative (between 0 and -0.01 per year) over western China and positive (between 0 and 0.4 per year) for most of the pixels over northern China and p-value was above 0.05 almost everywhere in those regions.

In P2 (Figs.7 and 8, middle panel), the AOD tendencies were opposite in sign, compared to P1. A statistically significant decrease in AOD of -0.02 to -0.04 per year (Fig.7, middle panel), or between -7% and -15% of AOD per year (Fig.8, middle panel) was observed in P2 over SE China. In Sichuan and Henan regions the AOD decrease was higher, reaching -0.1 per year. The highest

negative AOD tendency (ca. -15% of AOD per year) was observed in northern China over the east of Inner Mongolia area. Small AOD decrease was observed over the Taklamakan Desert; small AOD increase was observed over the most NE part of China. Over other areas, the AOD tendencies in P2 were close to zero.

The opposite in sign and similar in absolute values AOD tendencies in P1 and P2 partly cancel each other, when interannual AOD tendencies for the whole period are considered. Over most of the China, the AOD tendencies estimated for 1995-2017 are close to

zero. Slightly positive AOD tendencies (ca. 0.1, or 2% of AOD, per year) are observed in the BTH, Shandong and Henan regions.



### 5.2.2 Seasonal AOD tendencies over China.

Long-term AOD variations and their effects on the AOD seasonality over China have been discussed in Part I. Here we show the AOD seasonal tendencies, which are not always similar to the interannual tendencies.

In P1 (Figs. 9 and 10, left panel), the AOD increase of 0.1-0.5 (ca. 3-7% of AOD) per year in summer (JJA) and autumn (SON) contributed most to the annual AOD increase in P1 over SE China. In spring (MAM), the AOD tendency was positive over Yangzi River Delta (0.1-0.3, or 2-4% of AOD, per year), northern China and west of the Tibetan Plateau and negative over other parts of China. In winter (DJF), when the coverage of the satellite data is lower compared to other seasons (Part I), irregular patterns of both positive and negative AOD tendencies are observed over China.

In P2, the AOD decrease is observed in all seasons over the most part of China (Figs. 9 and 10, middle panel). The strongest AOD decrease (up to -0.16 per year) was observed over the northern part of SE China in summer. AOD decrease was a bit lower for other seasons over SE China, while for central and western China the AOD tendencies were close to zero in P2. The relative AOD tendency was between -5% and -10% per year in winter and spring over SE China and more than 15% per year over the Sichuan region in summer and autumn.

Similar to the annual AOD tendencies, the opposite AOD seasonal tendencies in P1 and P2 are partly cancelled by each other in WP (Figs. 9 and 10, right panel). The low AOD decrease (between -0.01 and -0.03 per year) has been observed in WP in spring in the Sichuan region. A small AOD increase (0.01-0.03 per year) was observed over BTH in winter and summer and over Henan in autumn. For the other parts of China, the AOD tendencies for the WP are close to zero. Small negative relative AOD tendencies (between -3% and -5% per year) were observed in Inner Mongolia in spring. Similar positive relative AOD tendencies (between 3% and 5% per year) were observed in the NE of China. For other seasons and areas, the AOD relative tendencies were close to zero. However, the linear fitting of the individual L3 pixels was not often providing statistically significant results (as indicated by lack of green dots for corresponding pixels) and the single pixel peaks in AOD tendencies are not considered here. The low significance of the tendencies can be explained by the low coverage (mostly in winter and spring over snow-covered areas, Part I), short length of the periods considered and high interannual variations. To reveal more detailed differences in AOD tendencies over China, we apply similar fitting to annual and seasonal AOD averaged over the selected regions of China.







**Figure 9.** AOD tendencies (per year, see colorbar) from seasonally aggregated combined AOD time series for three periods: 1995-2006 (P1), 2011-2017 (P2) and 1995-2017 (WP). Individual pixels are marked with the green dots when the linear fit was giving the statistically significant results.





**Figure 10. AOD relative tendencies (per year, see colorbar) from seasonally aggregated combined AOD time series for three periods: 1995-2006 (P1), 2011-2017 (P2) and 1995-2017 (WP).**



### 5.3 AOD tendencies for the selected regions

### 5.3.1 Annual AOD tendencies

In Fig. 11 we show the annual AOD combined time series (black line) for China, SE China and the 10 selected regions (corresponding red numbers in the left upper corner for each region), which are defined in Sect. 3. The linear fitting has been applied
to AOD for two periods, P1 and P2 (dashed lines). For P1 and P2, we show the AOD tendency (dAOD) per year. The AOD tendencies and results for linear fitting for P1, P2 and WP are summarised in Appendix, Table A1.

For both periods, the results for linear fitting were of high confidence (p-value<0.05), except for regions 8-10, where, as discussed in Part I (Table S1), ADV has low coverage and MODIS has difficulties in retrieving AOD over bright surfaces.

As expected, the AOD tendencies (dAOD)  and AOD relative tendencies were positive in P1 in all chosen regions, except for the
sparsely populated Tibetan Plateau (region 8), where AOD is very low and undergoes very little year-to-year variations. The maximum AOD increase in P1 (0.020, or 4% of AOD, per year) was observed in Shanghai area (region 2) and PRD and Guangxi province (region 7). In BTH (region 1), and Hunan-Guizhou (region 4), dAOD was also high (0.016, or 3% of AOD, per year). Those regions strongly contributed to the AOD increase in SE China (0.014, or 3% of AOD, per year).

In P2, the AOD decrease has been observed in all selected regions (1-10) and thus in SE China and all of China. In absolute numbers,
dAOD was almost twice higher in P2 than in P1. The most rapid AOD decrease (ca. -0.045, or -8% of AOD, per year) was observed in central regions of SE China (regions 4 and 6), while for the rest of SE China, including regions 1,2,3,5 and 7, dAOD was about -0.03, or ca -6% of AOD, per year. For regions 8-10, which are less populated and less industrialized, the dAOD was lower (-0.002, -0.014 and -0.004, respectively, or -2%, -5% and -1% per year).





**Figure 11. Annual AOD long-time series (black line) combined from ATSR and MODIS for China, SE China and 10 selected regions (correspondent red numbers in the left upper corner for each region). Results for linear fitting of AOD are shown for two periods, P1 (1995-2011) and P2 (2011-2017), marked with black arrows in upper left subplot. Fitting lines (dashed lines) and AOD tendencies (numbers) are shown in red, when AOD tendency was positive and in blue, when AOD tendency was negative. The corresponded relative AOD tendencies for each fit are shown in brackets (in black, %).**

## 5.3.2 Seasonal AOD tendencies

As shown in Part I, spatial and temporal patterns in the seasonal AOD differ from that of the annual AOD. To examine if the AOD tendencies in different seasons equally contributed to the annual AOD year-to-year changes or if the AOD year-to-year changes are more pronounced in certain seasons, we applied similar regression analysis for the seasonal aggregates of AOD for P1 and P2. As



for the annually averaged AOD (Fig.11), Fig.12 shows the seasonally combined AOD time series for China, SE China and the 10 selected regions. We also show the linear fitting of AOD for two periods, P1 and P2. For P1 and P2, we show the AOD tendency (dAOD) per year and the corresponding relative AOD tendencies. Time series, linear fitting line and the AOD tendency are shown in colours related to each season.

In P1, AOD does not have large temporal variability (within ±0.01 per year) in winter and spring in all regions in China (Fig.12). In winter, negative AOD tendency of -0.017 or -3% of AOD per year was observed in the Sichuan area (region 6) and in regions 9 and 10, where AOD coverage is low winter and high AOD variability is observed. Small negative AOD tendency (ca -0.006, or -2% of AOD, per year) is observed in the Hunan-Guizhou and PRD areas (regions 4 and 7) and small positive (ca 1-2% per year) in other regions. In spring, dAOD was slightly positive in regions 4, 8 and 10 and slightly negative in other regions. The AOD increase

was discovered in all regions (1-7) in eastern China in summer and autumn. In regions 1, 2, 5 and 6 (north and west of SE China region, respectively) AOD positive tendency was stronger in summer (ca 0.020, or 3-4% of AOD, per year). In regions 3, 4 and 7 (central east to southern part of SE China region) dAOD was stronger in autumn, increasing towards south and reaching maxima (0.032 or 6% of AOD per year) in region 7. Over whole SE China, dAOD was 0.020, or 4% of AOD, per year in summer and 0.016, or 4% of AOD, per year in autumn, while in winter and spring dAOD was close to zero from year to year. In western and northern

China (regions 8-10), year-to-year changes in the seasonal AOD in P1 were low (within ±0.01 per year). Over the whole China, AOD showed an increase of 0.008, or 2% of AOD, and 0.006, or 3% of AOD, per year in summer and autumn, respectively, and low decrease (ca. -0.02, or -1% of AOD, per year) in winter and spring. Thus, summer and spring contribute most to the annual AOD increase (Fig. 4) during years 1995-2006 (P1) and AOD changes were considerably higher in SE China compared to other areas of China. Note, that the p-value for linear fit applied here was often above 0.05 (Appendix, Table A2). However, similar AOD

tendencies in neighboring regions (with similar population density and economic activity and meteorological conditions) allow making the overall conclusions.

  In SE China, which includes regions 1-7, the strong AOD increase (0.020, or 4% of AOD, per year) was observed in summer. The second maxima was revealed in autumn (0.016, or 4% of AOD, per year). In winter, AOD increase was small (0.001 per year), while in spring a small AOD decrease (-0.003, or -1% of AOD, per year) has been observed.

AOD tendencies averaged seasonally over all of China for P1 show similar AOD increase in summer (0.008, or 2% of AOD, per year) and in autumn (0.006, or 3% of AOD, per year). In winter and spring, AOD was slightly decreasing (by - 1% of AOD, per year).



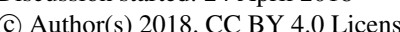


**Figure 12. AOD seasonal long-time series (solid lines), linear fitting (dashed lines) and AOD tendency (numbers) for P1 (1995-2011) and P2 (2011-2017) are shown in colours related to each season (blue for DJF, winter; purple for MAM, spring; green for JJA, summer; and light blue for SOA, autumn). The corresponded relative AOD tendencies for each fit are shown in brackets (in black, %).**

In P2, the AOD tendency was negative and about twice stronger (in absolute numbers) than in P1 over the selected regions. In winter, the AOD decrease was high (between -0.024 and -0.029 per year) in regions 1, 2, 5. The relative AOD tendency was of -11% per year in region 5, which is west of BTH area, between -4% and -6% per year over SE China and regions 1-7, and lower over other areas. In spring, AOD has been decreasing stronger compared to winter in regions 3, 4, 6 and 9. The high negative relative



tendency (from -6% to -9%) has been observed in regions 4-9 in spring. In summer, the high negative AOD relative tendency (between -10% and -12%) was observed in the southern part of SE China, regions 4-7. The highest absolute AOD decrease (-0.082, or -9% of AOD, per year) was observed in summer in BTH area (region 1). In autumn, AOD tendencies were lower compared to spring and summer in most of the regions. In SE China, which includes regions 1-7, the strongest AOD decrease (-0.048, or -9% of AOD, per year) was observed in summer. For other seasons, AOD was decreasing by -0.022, -0.028 and -0.023 per year (for DJF, MAM, SON, respectively), which is ca. -6% of AOD per year for each season.

Over all of China, the AOD decrease was also more pronounced in summer (-0.021, or -7% of AOD, per year). For other seasons, AOD was decreasing by -0.012, -0.014 and -0.011 per year (for DJF, MAM, SON, respectively), which is -4% of AOD per year in winter and spring and -5% of AOD per year in autumn.

Thus, the AOD changes in P2 in summer contribute most to the AOD annual year-to-year variability. AOD seasonal year-to-year changes are more pronounced over eastern China, which is explained by the uneven regional economic development in China.

As discussed above, the linear fitting of the whole period WP (1995-2017) does not give significant results, when applied to the annual AOD time series, since annual time series show positive AOD tendency in P1 and negative AOD tendency in P2 in almost all regions (except region 8, as discussed above). For seasonal aggregates, where dAOD has the same sign in P1 and P2 (e.g. spring time series in regions 6 and 7), linear fitting can be applied for the whole period. However, this behavior, where AOD tendency has similar sign in both P1 and P2, is an exception over China and thus, as for yearly aggregates, linear fitting results for the whole period for seasonal aggregates are shown in Table A2 but not discussed in detail.

## 6. Summary and conclusions

With the rapid economic development and further urbanization, anthropogenic and natural aerosols accumulate in China. However, the emission reduction policies in China have been changing during the last two decades, considerably strengthening the emission control after 2011. Here we investigate whether the tendencies in AOD in last two decades are related to emission changes as well as to the pollution control policies in China.

The limited lifetime of satellites makes it impossible to follow the AOD changes for several decades using time series from only one instrument. In this paper, we introduced a method to construct a combined AOD time series using the ATSR and MODIS/Terra sensors, which together cover the period of 1995-2017. The method is based on the ADV and MODIS comparison discussed in Part I. In brief, (1) ADV and MODIS show similar AOD annual and seasonal spatial and temporal patterns and (2) ADV is negatively biased, while MODIS is positively biased against AERONET AOD; in absolute values, the biases are similar to each other. The method was applied pixel-wise to annual and seasonal AOD aggregates from ADV and MODIS to construct the combined AOD time series.

The long-term (1995-2017) annual and seasonal combined AOD time series are presented for China, southeaster (SE) China and 10 selected regions. Linear regression has been applied to individual L3 (1°x1°) pixels of the annual and seasonal combined AOD time





series to estimate the AOD tendencies over China for three periods: 1995-2006 (P1) and 2011-2017 (P2), as regarding the changes in the emission reduction policies, and the whole period 1995-2017(WP), when the combined AOD time series is available. The main results and conclusions are summarised below.

- Positive tendencies of annual AOD (0.006, or 2% of AOD, per year) prevailed across all of mainland China before 2006
due to emission increases induced by rapid economic development. In P1, AOD was s increasing strongly over the wide industrial areas. Thus, in SE China, the annual AOD positive tendency in 1995-2006 was 0.014, or 3% of AOD, per year in SE China, reaching maxima (0.020, or 4% of AOD, per year) in Shanghai and the Pearl River Delta regains.

- Negative AOD tendencies (-0.015, or -6% of AOD, per year) were identified across most of China after 2011 in conjunction with effective emission reduction in anthropogenic primary aerosols, $SO_2$ and $NO_x$ (Jin et al., 2017, van der
A et al., 2017). The air quality mostly improved at the developed regions. Overall, AOD decrease in P2 was 2-3 times stronger than AOD increase in P1 over most of the SE China. The strongest AOD decrease in P2 is observed in the Chengdu (-0.045, or -8% of AOD, per year) and Zhengzhou (-0.046, or -9% of AOD, per year) areas, while over the North China plane and coastal areas the AOD decrease was <-0.03, or ca -6% per year. In the less populated areas, the AOD decrease was small.

- The AOD tendencies for the whole period 1995-2017 were much less pronounced. The reason for this is that positive AOD tendency has been observed at the beginning of WP (in P1) and negative AOD tendency has been observed at the end of WP (in P2), which partly cancel each other during 1995-2017.

- Seasonal patterns in the AOD regional long-term tendencies are evident. The contribution of seasonal AOD tendencies to annual tendencies was not equal along the year. While the annual AOD tendency was positive in P1, the AOD tendencies
in winter and spring were slightly negative (ca. -0.002, or -1% of AOD, per year) over the most of China during that period. AOD tendencies were positive in summer (0.008, or 2% of AOD, per year) and autumn (0.006, or 6% of AOD, per year) over all mainland China and SE China (0.020, or 4% of AOD, per year and 0.016, or 4% of AOD, per year in summer and autumn, respectively). As in P1, the AOD negative tendencies in P2 were higher compared to other seasons in summer over China (ca. -0.021, or -7% of AOD, per year) and over SE China (ca. -0.048, or -9% of AOD, per year). In the east,
seasonal variations in AOD tendencies were less pronounced.

Thus, in the current study the effect of the changes in the emission regulations policy in China is evident in AOD decrease after 2011. The effect is more visible in the highly populated and industrialized regions in SE China.

**Data availability**

The ATSR data used in this manuscript are publicly available (after registration a password will be issued) at: http://www.icare.univ-lille1.fr/. MODIS data are publicly available at: https://ladsweb.modaps.eosdis.nasa.gov/. AERONET data are available at AERONET: https://aeronet.gsfc.nasa.gov/



**Acknowledgements**:

Work presented in this contribution was undertaken as part of the MarcoPolo project supported by the EU, FP7 SPACE Grant agreement no. 606953 and as part of the Globemission project ESA-ESRIN Data Users Element (DUE), project AO/1-6721/11/I-NB, and contributes to the ESA/MOST DRAGON4 program. The ATSR algorithm (ADV/ASV) used in this work is improved with support from ESA as part of the Climate Change Initiative (CCI) project Aerosol_cci (ESA-ESRIN projects AO/1-6207/09/I-LG and ESRIN/400010987 4/14/1-NB). Further support was received from the Centre of Excellence in Atmospheric Science funded by the Finnish Academy of Sciences Excellence (project no. 272041). Many thanks are expressed to NASA Goddard Space Flight Center (GSFC) Level 1 and Atmosphere Archive and Distribution System (LAADS) (http://ladsweb.nascom.nasa.gov) for making available the L3 MODIS/Terra C6.1 aerosol data. The AERONET team is acknowledged for establishing and maintaining the AERONET sites used in this study.

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



**Appendix**

**Table A1. AOD tendency (dAOD) per year and statistics for the annual combined AOD time series linear fitting (p-value, bias, slope and absolute error (ae, %)) for three periods: P1 (1995-2006), P2 (2011-2017) and WP (whole period, 1995-2017) for different regions (r, where region 11 is the whole mainland China and region 12 is mainland SE China).**

| r | dAOD | P1 fit | | | | dAOD | P2 fit | | | | dAOD | WP fit | | | |
|---|------|---------|------|-------|------|------|---------|------|-------|------|------|---------|------|-------|------|
| | | p-value | bias | slope | ae,% | | p-value | bias | slope | ae,% | | p-value | bias | slope | ae,% |
| 1 | 0,018 | 0,002 | -35,39 | 0,018 | 6,82 | -0,038 | 0,003 | 77,62 | -0,038 | 5,65 | 0,008 | 0,004 | -15,58 | 0,008 | 6,95 |
| 2 | 0,020 | 0,001 | -38,63 | 0,020 | 6,69 | -0,033 | 0,008 | 67,28 | -0,033 | 7,23 | 0,005 | 0,049 | -10,02 | 0,005 | 7,79 |
| 3 | 0,010 | 0,015 | -20,13 | 0,010 | 7,45 | -0,029 | 0,010 | 57,76 | -0,029 | 9,15 | 0,000 | 0,833 | -0,39 | 0,000 | 7,79 |
| 4 | 0,018 | 0,000 | -35,49 | 0,018 | 6,41 | -0,045 | 0,001 | 91,59 | -0,045 | 6,72 | 0,003 | 0,277 | -5,46 | 0,003 | 8,88 |
| 5 | 0,012 | 0,013 | -24,46 | 0,012 | 10,33 | -0,036 | 0,000 | 73,50 | -0,036 | 3,80 | 0,002 | 0,291 | -4,15 | 0,002 | 9,64 |
| 6 | 0,012 | 0,043 | -23,72 | 0,012 | 8,63 | -0,046 | 0,000 | 93,19 | -0,046 | 4,64 | -0,004 | 0,192 | 7,94 | -0,004 | 8,84 |
| 7 | 0,020 | 0,001 | -39,27 | 0,020 | 7,13 | -0,032 | 0,006 | 64,22 | -0,032 | 6,96 | 0,004 | 0,159 | -6,62 | 0,004 | 8,02 |
| 8 | -0,001 | 0,427 | 1,49 | -0,001 | 7,41 | -0,002 | 0,216 | 4,87 | -0,002 | 9,07 | -0,001 | 0,086 | 1,14 | -0,001 | 5,02 |
| 9 | 0,010 | 0,072 | -19,68 | 0,010 | 17,79 | -0,014 | 0,055 | 29,37 | -0,014 | 11,31 | 0,002 | 0,184 | -4,49 | 0,002 | 11,20 |
| 10 | 0,007 | 0,172 | -14,49 | 0,007 | 17,21 | -0,004 | 0,476 | 8,87 | -0,004 | 10,02 | 0,002 | 0,153 | -4,36 | 0,002 | 9,91 |
| 11 | 0,006 | 0,006 | -11,61 | 0,006 | 6,00 | -0,015 | 0,001 | 30,20 | -0,015 | 4,39 | 0,001 | 0,397 | -1,32 | 0,001 | 6,12 |
| 12 | 0,014 | 0,000 | -27,28 | 0,014 | 4,81 | -0,033 | 0,000 | 66,84 | -0,033 | 3,60 | 0,002 | 0,259 | -4,05 | 0,002 | 7,51 |

**Table A2. AOD tendency (dAOD) per year and statistics for linear fitting (p-value, bias, slope, absolute error (ae, %)) for the seasonally combined AOD time series for three periods: P1 (1995-2006), P2 (2011-2017) and WP (whole period, 1995-2017) for different regions (r, where region 11 is the whole mainland China and region 12 is the mainland SE China) and different seasons (s, 1 – winter, 2 – spring, 3 – summer, 4 - autumn).**

| r | s | dAOD | P1 fit | | | | dAOD | P2 fit | | | | dAOD | WP fit | | | |
|---|---|------|---------|------|-------|------|------|---------|------|-------|------|------|---------|------|-------|------|
| | | | p-value | bias | slope | ae,% | | p-value | bias | slope | ae,% | | p-value | bias | slope | ae,% |
| 1 | 1 | 0,012 | 0,150 | -24,1 | 0,012 | 17,5 | -0,024 | 0,213 | 48,8 | -0,024 | 16,9 | 0,011 | 0,001 | -21,9 | 0,011 | 10,8 |
| 1 | 2 | -0,002 | 0,791 | 5,1 | -0,002 | 9,7 | -0,018 | 0,261 | 36,3 | -0,018 | 12,4 | -0,003 | 0,218 | 7,4 | -0,003 | 6,8 |
| 1 | 3 | 0,024 | 0,013 | -47,0 | 0,024 | 9,4 | -0,082 | 0,017 | 166,0 | -0,082 | 14,0 | 0,011 | 0,045 | -20,8 | 0,011 | 10,5 |




| | | | | | | | | | | | | | | | | |
|---|---|---|---|---|---|---|---|---|---|---|---|---|---|---|---|---|
| 1 | 4 | 0,015 | 0,049 | -28,5 | 0,015 | 12,9 | -0,014 | 0,166 | 28,0 | -0,014 | 8,6 | 0,006 | 0,015 | -11,6 | 0,006 | 8,1 |
| 2 | 1 | 0,001 | 0,819 | -1,4 | 0,001 | 7,7 | -0,027 | 0,085 | 54,1 | -0,027 | 13,6 | 0,004 | 0,086 | -6,8 | 0,004 | 7,4 |
| 2 | 2 | 0,005 | 0,493 | -9,5 | 0,005 | 7,9 | -0,025 | 0,034 | 50,9 | -0,025 | 7,2 | -0,003 | 0,272 | 6,5 | -0,003 | 6,1 |
| 2 | 3 | 0,026 | 0,007 | -51,7 | 0,026 | 12,1 | -0,043 | 0,190 | 86,8 | -0,043 | 23,3 | 0,008 | 0,086 | -15,7 | 0,008 | 12,3 |
| 2 | 4 | 0,019 | 0,010 | -36,6 | 0,019 | 11,3 | -0,027 | 0,003 | 54,5 | -0,027 | 5,8 | 0,003 | 0,339 | -4,8 | 0,003 | 9,6 |
| 3 | 1 | 0,001 | 0,779 | -1,6 | 0,001 | 7,5 | -0,023 | 0,049 | 47,0 | -0,023 | 13,0 | -0,001 | 0,694 | 1,7 | -0,001 | 7,3 |
| 3 | 2 | -0,010 | 0,153 | 19,9 | -0,010 | 8,4 | -0,032 | 0,028 | 64,0 | -0,032 | 10,1 | -0,003 | 0,240 | 6,3 | -0,003 | 6,8 |
| 3 | 3 | 0,012 | 0,039 | -22,7 | 0,012 | 10,5 | -0,029 | 0,157 | 59,6 | -0,029 | 23,9 | 0,000 | 0,870 | 1,3 | 0,000 | 10,9 |
| 3 | 4 | 0,017 | 0,016 | -33,8 | 0,017 | 13,8 | -0,028 | 0,005 | 56,6 | -0,028 | 9,1 | -0,001 | 0,861 | 1,3 | -0,001 | 11,6 |
| 4 | 1 | -0,008 | 0,245 | 15,4 | -0,008 | 11,8 | -0,019 | 0,093 | 38,3 | -0,019 | 11,8 | -0,002 | 0,415 | 4,3 | -0,002 | 8,8 |
| 4 | 2 | 0,009 | 0,337 | -16,6 | 0,009 | 10,2 | -0,045 | 0,005 | 90,5 | -0,045 | 8,8 | -0,004 | 0,245 | 7,9 | -0,004 | 7,9 |
| 4 | 3 | 0,017 | 0,012 | -32,4 | 0,017 | 9,9 | -0,057 | 0,076 | 115,5 | -0,057 | 27,0 | 0,004 | 0,357 | -6,9 | 0,004 | 12,8 |
| 4 | 4 | 0,026 | 0,000 | -51,7 | 0,026 | 9,8 | -0,040 | 0,000 | 80,8 | -0,040 | 5,4 | 0,004 | 0,279 | -6,5 | 0,004 | 11,0 |
| 5 | 1 | 0,000 | 0,947 | -0,5 | 0,000 | 18,6 | -0,029 | 0,004 | 58,3 | -0,029 | 12,9 | 0,001 | 0,775 | -0,9 | 0,001 | 13,0 |
| 5 | 2 | -0,003 | 0,530 | 7,1 | -0,003 | 9,6 | -0,025 | 0,017 | 51,0 | -0,025 | 11,2 | -0,005 | 0,007 | 10,5 | -0,005 | 7,0 |
| 5 | 3 | 0,020 | 0,040 | -38,7 | 0,020 | 15,3 | -0,057 | 0,009 | 115,0 | -0,057 | 13,6 | 0,005 | 0,215 | -9,5 | 0,005 | 12,7 |
| 5 | 4 | 0,010 | 0,010 | -19,9 | 0,010 | 10,5 | -0,017 | 0,013 | 34,4 | -0,017 | 7,6 | 0,003 | 0,041 | -5,5 | 0,003 | 7,9 |
| 6 | 1 | -0,017 | 0,161 | 33,8 | -0,017 | 18,8 | -0,026 | 0,298 | 52,7 | -0,026 | 27,3 | -0,005 | 0,251 | 10,5 | -0,005 | 14,4 |
| 6 | 2 | -0,019 | 0,182 | 39,4 | -0,019 | 14,6 | -0,048 | 0,003 | 97,2 | -0,048 | 9,0 | -0,014 | 0,001 | 29,1 | -0,014 | 9,3 |
| 6 | 3 | 0,019 | 0,039 | -38,2 | 0,019 | 14,2 | -0,060 | 0,000 | 120,5 | -0,060 | 7,2 | -0,002 | 0,641 | 4,0 | -0,002 | 12,2 |
| 6 | 4 | 0,012 | 0,047 | -22,9 | 0,012 | 9,6 | -0,032 | 0,024 | 65,5 | -0,032 | 13,5 | -0,004 | 0,146 | 8,4 | -0,004 | 9,7 |
| 7 | 1 | -0,005 | 0,471 | 11,1 | -0,005 | 12,4 | -0,019 | 0,232 | 37,7 | -0,019 | 16,8 | -0,003 | 0,219 | 7,0 | -0,003 | 9,1 |
| 7 | 2 | -0,006 | 0,395 | 11,9 | -0,006 | 7,5 | -0,025 | 0,231 | 51,4 | -0,025 | 13,1 | 0,007 | 0,040 | -13,7 | 0,007 | 7,5 |
| 7 | 3 | 0,026 | 0,012 | -51,6 | 0,026 | 16,0 | -0,033 | 0,103 | 66,7 | -0,033 | 18,6 | 0,001 | 0,703 | -2,3 | 0,001 | 12,7 |
| 7 | 4 | 0,032 | 0,001 | -63,6 | 0,032 | 12,8 | -0,035 | 0,008 | 70,6 | -0,035 | 9,7 | 0,002 | 0,690 | -2,5 | 0,002 | 13,1 |
| 8 | 1 | 0,002 | 0,123 | -4,0 | 0,002 | 14,0 | 0,001 | 0,657 | -1,3 | 0,001 | 12,7 | 0,000 | 0,942 | 0,1 | 0,000 | 9,3 |
| 8 | 2 | -0,001 | 0,551 | 1,8 | -0,001 | 6,7 | -0,001 | 0,505 | 1,8 | -0,001 | 4,1 | 0,000 | 0,918 | 0,1 | 0,000 | 5,1 |
| 8 | 3 | -0,001 | 0,377 | 1,5 | -0,001 | 6,3 | 0,001 | 0,888 | -0,9 | 0,001 | 15,5 | 0,000 | 0,488 | -0,4 | 0,000 | 5,6 |
| 8 | 4 | -0,003 | 0,016 | 5,4 | -0,003 | 10,0 | -0,003 | 0,309 | 5,2 | -0,003 | 16,4 | -0,001 | 0,002 | 2,6 | -0,001 | 7,8 |
| 9 | 1 | -0,039 | 0,251 | 69,5 | -0,035 | 115,3 | 0,001 | 0,917 | -1,6 | 0,001 | 30,1 | -0,011 | 0,101 | 22,0 | -0,011 | 63,9 |
| 9 | 2 | -0,047 | 0,132 | 94,0 | -0,047 | 39,4 | -0,029 | 0,065 | 58,3 | -0,029 | 14,2 | -0,014 | 0,058 | 27,9 | -0,014 | 20,9 |
| 9 | 3 | 0,007 | 0,432 | -14,5 | 0,007 | 28,6 | -0,009 | 0,505 | 18,9 | -0,009 | 25,6 | 0,000 | 0,954 | 0,6 | 0,000 | 16,6 |
| 9 | 4 | 0,002 | 0,403 | -4,5 | 0,002 | 18,7 | -0,013 | 0,127 | 26,4 | -0,013 | 23,8 | 0,002 | 0,060 | -4,3 | 0,002 | 13,7 |
| 10 | 1 | -0,028 | 0,200 | 49,8 | -0,025 | 68,3 | -0,013 | 0,337 | 27,0 | -0,013 | 33,6 | -0,006 | 0,236 | 10,8 | -0,005 | 33,5 |
| 10 | 2 | 0,004 | 0,793 | -6,7 | 0,004 | 28,8 | 0,001 | 0,852 | -2,2 | 0,001 | 11,3 | -0,003 | 0,405 | 5,8 | -0,003 | 15,7 |
| 10 | 3 | 0,003 | 0,544 | -6,2 | 0,003 | 15,0 | -0,016 | 0,232 | 33,3 | -0,016 | 18,2 | 0,002 | 0,272 | -3,9 | 0,002 | 10,1 |
| 10 | 4 | 0,005 | 0,264 | -8,9 | 0,005 | 21,4 | -0,003 | 0,349 | 5,8 | -0,003 | 6,6 | 0,004 | 0,003 | -7,1 | 0,004 | 10,6 |
| 11 | 1 | -0,001 | 0,556 | 2,9 | -0,001 | 7,4 | -0,012 | 0,041 | 23,9 | -0,012 | 8,9 | 0,001 | 0,606 | -0,7 | 0,001 | 6,0 |
| 11 | 2 | -0,003 | 0,555 | 6,5 | -0,003 | 10,5 | -0,014 | 0,030 | 28,7 | -0,014 | 7,8 | -0,003 | 0,020 | 7,2 | -0,003 | 6,2 |
| 11 | 3 | 0,008 | 0,006 | -14,7 | 0,008 | 6,6 | -0,021 | 0,036 | 42,2 | -0,021 | 12,9 | 0,001 | 0,477 | -1,7 | 0,001 | 7,9 |
| 11 | 4 | 0,006 | 0,001 | -11,5 | 0,006 | 6,1 | -0,011 | 0,007 | 21,8 | -0,011 | 6,1 | 0,001 | 0,130 | -2,1 | 0,001 | 6,2 |
| 12 | 1 | 0,001 | 0,642 | -1,9 | 0,001 | 5,6 | -0,022 | 0,024 | 44,1 | -0,022 | 9,7 | 0,002 | 0,232 | -3,2 | 0,002 | 6,6 |
| 12 | 2 | -0,003 | 0,476 | 7,0 | -0,003 | 6,2 | -0,028 | 0,012 | 55,9 | -0,028 | 7,9 | -0,004 | 0,031 | 8,7 | -0,004 | 5,4 |
| 12 | 3 | 0,020 | 0,001 | -39,0 | 0,020 | 7,5 | -0,048 | 0,012 | 97,1 | -0,048 | 13,2 | 0,004 | 0,232 | -7,3 | 0,004 | 10,6 |
| 12 | 4 | 0,016 | 0,000 | -31,5 | 0,016 | 7,4 | -0,023 | 0,000 | 47,2 | -0,023 | 4,0 | 0,002 | 0,346 | -3,2 | 0,002 | 8,4 |