# Peer review of "Spatial and seasonal variations of aerosols over China from two decades of multi-satellite observations. Part 2: AOD time series for 1995-2017 combined from ATSR ADV and MODIS C6.1 and AOD tendencies estimation"

_Atmospheric Chemistry and Physics, 2018_

## Referee Comment (RC1) · Anonymous Referee #2 · 11 May 2018

My main concerns with this paper is the small amount of detail included both on which precisely datasets were used but also on presenting the actual findings. Even though the authors recognise three distinct periods of AOD behaviour in their timeseries, an increase, a plateau and a decrease, they present a number of figures and tables with linear trends on the entire period, which is statistically incorrect. I suggest they re-think this finding accordingly.

[Figure]

Furthermore, the two datasets, [A]ATSR and MODIS have a clear bias which, although discussed, has not sufficiently been excluded as the reason for the three periods identified. The tendencies calculated appear to be of the same order of magnitude as this bias which leads to the thinking that the bias might be responsible. Consider de-seasonalising the datasets before any further analysis is performed.

The constant reference to Part I is also rather tiring and I consider that a few lines of the findings of that Part would go greatly towards improving their comments.

The figures are not of optimal quality, I suggest increasing the pixel quality, and provide ACP with their *eps versions.

The topic is of high interest and merit but I find a number of changes will have to be made on the presentation of the datasets as well as of the findings.

Refer to the annotated document for further technical, wording as well as scientific comments.

Please also note the supplement to this comment:
https://www.atmos-chem-phys-discuss.net/acp-2018-288/acp-2018-288-RC1-supplement.pdf

———————————————————

[Figure]

**Supplement:**

[revised manuscript text omitted]

---

## Referee Comment (RC2) · A. M. Sayer (Referee) · 25 May 2018

Summary:

I am writing this review under my own name (Andrew Sayer) as I have previously discussed this research with the authors, and am on the team responsible for the MODIS aerosol data products being used in the study. I also reviewed the paper de Leeuw et al (2018), which is in some sense a predecessor to this study, and the Part I of this

paper also by Sogacheva et al and also currently in ACPD. I feel I am able to provide an impartial review, but am signing the review in the interests of transparency.

The goal of this pair of papers is to look at spatial and temporal (seasonal/interannual) variations of AOD over China. This is accomplished mainly by using two satellite data sets: the ADV algorithm applied to the combined ATSR2/AATSR record (1995-2012), and the combined Deep Blue/Dark Target algorithms applied to the MODIS Terra record (2000 onwards) from the latest Collection 6.1. Part I contains some validation results and an initial look at the time series, while this Part II focuses on trends ("tendencies" in the authors' terminology) during several time periods where emissions policies may have influenced the aerosol loading. These papers are linked so I will summarize my review of Part I first (which be found on the ACPD page at https://www.atmos-chem-phys-discuss.net/acp-2018-287/ ), since this Part II requires Part I to stand on. For Part I, I have recommended revisions and re-review. The two main technical threads of my review of Part I were that (1) more needed to be done to establish the validity of treating ATSR2 and AATSR as a single record (which is the underlying but untested assumption), and (2) some of the time series analysis in Part I should be moved to Part II to keep the flow of both papers better and avoid some redundancy. So this review should be read with that in mind.

My overall recommendation for this Part II is also for major revisions and re-review. It's an interesting and important topic, but I don't think it is ready for publication in current form. I would like to review the revision; I would prefer if Part I can be revised and eventually accepted for publication first, if possible, so that we have that as a stable version to refer back to when reviewing a revision of this Part II, since the papers are quite closely linked. This is an interesting study but I think (see below) that the ATSR/MODIS merging technique requires some more examination, and also the conclusions would be better supported by including additional meteorological and/or geophysical data products in the analysis (so we can see whether AOD changes are likely to be the result of policy, or whether weather patterns may be an influence here).

[Figure]

Uncertainties in the method and results also need better quantification. Note I am not an expert on policy or emissions, so my comments mostly focus on the statistics and AOD data. Hopefully another reviewer can comment on policy/emissions in more detail – my lack of comments is due to a lack of expertise to judge in those areas.

The quality of language is overall good and any issues can probably be dealt with by Copernicus' copy-editing and typesetting process. Therefore my review mainly concentrates on technical abstracts. I have tried to separate each main comment into its own paragraph to respond to. Here, PXLY refers to page X, line Y.

Specific comments:

Abstract: I would condense this into one paragraph if possible and shorten it to highlight the main findings. For example I think the authors can cut out the discussion on linear trends across the whole period, since one of the main points of the study is that it should not be considered one period due to the changes in emissions policy. I'd also cut out the discussion of annual trends/tendencies since I think (as discussed before, and below) only seasonal trends are meaningful due to the seasonal differences in aerosol loading, type, and retrieval coverage (from e.g. cloud cover, snow) aliasing into the annual means in a complicated way. Also, the papers cited in the abstract can be removed – these citations are in the body text, they are just adding length here; traditionally one doesn't need to provide citations to back up statements in the abstract because that's what the rest of the paper is for.

P3L2-3: I would avoid giving urls like this as citations here, particularly since the latter is an opinion piece. Urls are not always stable and one can't be sure the content is going to change or is valid. It would be better to cite something with a DOI or official publication number. For example the first link is for the World Bank so there must be some report or something which can be used.

Figure 1: Likewise, I would not give an url here for the population data used. If you click through the url, it gives a citation for the data set which should be used instead.

A couple of other things jump out at me from this figure. First, it seems that the largest population change in this region is not in fact China, but India. If population acts as a driver for anthropogenic aerosol emissions, one might expect that observed aerosol changes in China may be influenced by changes in transported aerosols from India. If this contribution cannot be quantified, it means that one cannot state that observed changes in China are a result of changes in Chinese policy. (The fact that aerosols don't follow national borders is one reason why in general I prefer regional studies to national studies – you have to be able to account for the broader context of regional emissions/meteorology changes.) Secondly, it looks like the population in the Sichuan Basin area (30 N, 105 E) has dropped somewhat since 2000, while the rest of China has been flat or steadily increasing. Is this right? I did a quick search online and it looks like the Chengdu metropolitan area population is increasing (http://worldpopulationreview.com/world-cities/chengdu-population/ - perhaps this is the red dot on the map here – but the overall population of Sichuan province is fairly stable (http://population.city/china/adm/sichuan/ ). However I'm not sure of the reliability of these sources, and it is difficult to estimate the total population from these maps of population density because the colour scales seem to saturate and we don't have grid size information. So perhaps people in Sichuan province are becoming more concentrated in Chengdu, I don't know. The point is, I think both of these aspects should be discussed in more detail in the manuscript.

Section 2: This is largely a repetition of Part I, in that it is introducing the satellite data used. I understand a recap of the data is needed, but I think that this could be shortened. For example we don't need to know the spectral and spatial resolution of the ATSR and MODIS instruments, or provide the validation summary table. I think it's enough to say you're using the level 3 monthly products at 1 degree, and refer back to Part I for more details. Particularly since the number of 3-way matchups (AERONET, ATSR, and MODIS all together) was low and largely confined to cities in Eastern China, I think discussing these statistics in detail gives a (possibly false) impression that we can be confident that these are representative of relative performance across the whole

of China. Plus, instantaneous validation differences will not necessarily reflect differences in the monthly or longer means, and the method in Section 4 is meant to show and reconcile these differences. So I think Section 2 can be shortened to a couple of paragraphs.

Section 4, general: the large number of long subscripts on variable names makes things cumbersome to read (hard to visually follow the equations). I suggest replacing "ADV" subscripts with "A" and "MODIS" or "MOD" (both are used, but I think mean the same thing) subscripts with "M". Other subscript shortenings could include "y" for "year", "c" for "comb", or "rc" for "rel_corr", for example. This will make it easier to follow the equations.

My understanding of the method is that the overlapping ATSR/MODIS period (2000-2011) just averages the two time series. Then the size of this adjustment in the overlapping period is used to scale the pre-2000 ATSR data, and the post-2011 MODIS data, to generate the "merged" time series. This is done twice: once for an "annual" correction, used for the merged multiannual time series, and once for a set of four "seasonal" corrections, used for the merged seasonal time series. Further, the calculation is performed separately for each 1 degree grid cell. Is that a correct description? If so I would include a couple of sentences to that effect somewhere up the top, before the equations, in case the reader gets lost.

P10L27-28: the authors mention Bourassa et al (2014) as an example of this technique being used elsewhere. However I'm not sure that is necessarily a good justification. Bourassa et al were looking at stratospheric ozone, which has (to my knowledge) a somewhat longer lifetime and smoother spatial distribution than AOD, as well as fewer contextual (i.e. surface cover or type-dependent) uncertainties. These are very different error characteristics. Looking through the Bourassa paper, the relative differences between the ozone data sets used were in many cases a lot smaller than the differences seen for AOD here. So the method which is justifiable for one geophysical data set is not necessarily a good choice for another one. Are there other example applications, or justifications which can be made?

I have some more general concerns about this method for combining the ATSR and MODIS records, which become more severe because the purpose is trend analysis, which becomes particularly sensitive to values at the start and end points. Both the start and end points of the combined record are being adjusted by this method, and the "combined" to "adjusted MODIS" changeover also takes place at the same time (year 2011) as the start of the trend analysis period "P2" (2011-2017). So uncertainties in the merging process will unfortunately affect the trend analysis at the points at which a trend calculation is most susceptible to artefacts. I appreciate the authors' efforts to merge the ATSR and MODIS records here, and I don't know that there is any well-defined most appropriate way to do so. So this is one attempt which seems like a reasonable thing to try, and I am not sure what other way to harmonise the AOD record to suggest. However the fact remains that this method will introduce uncertainties, which may be systematic, and influence calculated trends. So, at a minimum, the possible influence of these effects must be quantified. Here are some thoughts about how to do that. The decision to do a simple average for the overlapping (2000-2011) period seems to have been made on the fact that the mean bias of the two data sets vs. AERONET, based on the limited available number of samples and limited available locations of samples, was roughly equal and opposite (so averaging might be expected to cancel out the bias on an average basis). Despite the fact that this was presented as a difference in ABSOLUTE AOD bias, the correction term is applied as a RELATIVE AOD correction. So that feels somewhat inconsistent. The authors could make maps of MODIS-ATSR on a monthly basis, both in absolute and relative terms, and see what it looks like. If the bias vs. AERONET of roughly 0.06 (for MODIS) and -0.07 (for ADV) is representative everywhere, then I would expect these maps of absolute difference to hover around values of 0.13 and have little spatial or temporal variability. If they don't, then it tells you that the AERONET matchups aren't representative of the bigger picture. In contrast if maps of relative AOD show small variability, it suggests that a relative scaling is more appropriate.

For the period with both ATSR and MODIS data (2000-2011), my understanding is that the two time series are averaged, and it is the correction across this 11-year period which is the basis for correction periods T1 and T3. So one other thing which could be tested is to see what the variability in the MODIS-ATSR difference is over 2000-2011, e.g. what the standard deviation of the difference is. If it is 0 then MODIS and ATSR are always offset the same amount. In reality it will be nonzero, and this additional variability should be propagated into the trend uncertainties discussed later in the paper. Although you might get a trend with apparently low noise, if you know that part of your time series may have uncertainties which aren't captured by this error model, then those errors (in this case, the interannual variability in MODIS-ATSR AOD in 2001-2011) should be added on when estimating the total uncertainty on a trend. A similar point can be made (see my review of Part I) when considering the combination of ATSR2 and AATSR to give the single combined ATSR record used as the basis here.

My remaining comments are more general, because I think that the above comments and discussion, plus the review of Part I, may necessitate a rewrite of some of the later sections of this study. Some of the points which I think need to be discussed here in more detail include:

1. Trend terminology. The authors say "tendency" rather than "trend" throughout. I prefer the more standard "trend". From previous discussions with the authors, my understanding is that they preferred the term "tendency" because they feel that "trend" should refer to a longer time period than considered here. My personal feeling is that "trend" is clearer for the reader, so long as it is made explicit that one should not extrapolate out of the time period under consideration.

2. Annual and whole period trends. I continue to think that, since the time series show seasonal variation and are not linear, it is not sensible to do whole period (WP) trends, or annual trends. I think it's best to show only the piecewise trends.

3. Trend breakpoints. The authors split the trend analysis into WP, an early P1, and

a late P2. These split times are informed by times where policy changes may have had an influence. However there also exist established statistical methods to estimate whether there are breakpoints in trends, and when these breakpoints are. It would be good to use these methods to see whether in fact any such breakpoints are detectable at the point the authors assume they are there.

4. Trend significance. The actual fitting mechanism and uncertainty estimation is not discussed in detail. P21L5 says it is linear regression (I assume ordinary least squares) and they define $p<0.05$ as statistically significant (I assume here this is defined as the estimated trend is at least twice as large as the estimated uncertainty in that trend, i.e. 2 sigma). Is autocorrelation considered? Often in trend studies lag 1 autocorrelation is estimated and used to correct the uncertainty estimate (because many geophysical data fields can be autocorrelated on time scales of months, seasons, or years). This gives a more realistic, and generally larger, estimate of the uncertainty on the trend because autocorrelation tends to be positive. See e.g. Weatherhead et al (JGR 1998, https://agupubs.onlinelibrary.wiley.com/doi/pdf/10.1029/98JD00995 ). If this is not done then there should be some evidence given why this model is reasonable. Further, as noted above, the uncertainty from the ATSR/MODIS merging exercise should ideally be considered. Additionally, there's the problem that since so many comparisons are being performed there are likely to be a number of false positives (trends which appear significant but are coincidental). See e.g. Wilks (JAMC 2006, https://journals.ametsoc.org/doi/10.1175/JAM2404.1 ) for more on the false discovery rate and how to deal with this sort of thing.

5. Presentation of trend uncertainties. In the later Figures and discussion, trends are given in terms of both absolute and relative AOD (where I guess relative AOD is defined with respect to some base year – this is unclear – since as the AOD changes by some absolute amount per year, the relative trend would change while the absolute trend would not). For clarity and comparability between regions, I would rather just see absolute AOD trends. I would also like to have the uncertainties estimated on the

trends be presented and discussed (the Appendix tables say they have absolute error in percent, but it's not clear to me what exactly that means, and it is more relatable to have in absolute units like the trend itself). For example at face value the sign of a trend may have changed between two time periods, but it may be that the difference is within the uncertainties of the individual trends, in which case it is more accurate to say that a change in behaviour cannot be identified. Thresholds on p-value are somewhat arbitrary and so I think it is generally more useful to present trend uncertainties instead (or in addition) in the text and tables in the Appendix.

6. Evidence of attribution for trends. It makes sense that changes in policy could affect emissions and change the AOD. However another key factor, which is examined in several other studies in China and elsewhere, is changes in meteorology. For example if there have been changes in air stagnation frequency, or aerosol transport pathways (for sources outside China as well as those inside China), then these might be magnifying or masking any trends resulting from policy changes. Additional data sets that might be able to support this would include emissions data bases, other satellite products (e.g. SO2 and NOx, which are briefly discussed), and meteorological reanalyses. Some of this has been done by other studies, and I'd like the discussion to go into more detail on those. But some of this may not have been and the authors might need to do these analyses themselves. Otherwise it is premature to try to state the reason for the trends.

In summary, the main thread is I feel it is important to be thorough and give reasonable uncertainties – say what we can – than to make a conclusion which isn't fully supported. Particularly since this is an inherently political topic. Maybe the data we have are not enough to be conclusive yet, in which case it is even more important to say as much as we can but no more and be clear about what the biggest uncertainties which we need to reduce to be able to answer the question are. The last sentences of the paper (P26L26-27: "Thus, in the current study the effect of the changes in the emission regulations policy in China is evident in AOD decrease after 2011. The effect is more

visible in the highly populated and industrialized regions in SE China.") are very strong statements, and this might indeed be the case. But I don't think the discussion of uncertainties is quite thorough enough, or other explanations and their contributions examined in enough detail, to make this case.

---

## Author Comment (AC1) · 6 Sep 2018

**Dear Referee#2,**

Thank you for your criticisms and suggestions to the manuscript. Most of the modifications suggested were considered in the manuscript and below are the comments to your questions/suggestions, which are now marked with **bold**.

**P1C1. Please add which AOD this refers to [I assume 550nm?] and then for the rest of the text you can keep it simply as AOD.**

- Wavelength is specified

**P1C2.**

- The preposition is removed

**P1C3. This comment applies to all such numbers given in the entire text: I fail to understand how AOD might increase by 2% per annum, i.e. 20% per decade. A change of 0.006 in AOD per annum cannot represent 2% of the total AOD.**
**Please clarify exactly what you mean and alter accordingly in all locations in the text.**
**Also, it is vital that you add an error estimate in this trend. This estimate is provided easily by all major languages. Since this paper is based on the trend analysis, a simple error estimate on this trend has to appear. At least.**

- If averaged for the tested period AOD is 0.3, then the annual increase of 0.006 is 2% of the mean AOD. In 10 years, AOD increase is 0.06, which is 20% of mean AOD. That means, that at the beginning of the period, AOD was ~ 0.27, while by the end of the period AOD was 0.33

**P1C4. This seems enormous... please check and alter.**

- Please, see my explanation above

**P2C1. ca? please use full word.**

- Replaces by "approximately"

**P2C2. This is another major issue in my opinion, which has to be addressed centrally and will possibly alter major parts of the rest of the paper. How do you justify a linear trend on a time series that you have shown is made up of a clear positive trend, a plateau and a negative trend? the linear analysis on the entire series has absolutely no meaning. I suggest you rethink the focus of the paper and alter the entire text accordingly.**

- Discussion on the whole period anomalies is removed from the manuscript.

**P2C3. Re-write.**

- See reply to P2C2

**P2C4. as expected.**

- Added

**P2C5. A further issue I have with this analysis is why did you not deseasonalise the time series before performing the linear trends calculation, as is statistically appropriate. Please comment both here as well as in the text.**

- Deseasonalisation was not needed since yearly averages were used to estimate inter-annual trend and seasonal aggregates for seasonal trends.

**P2C6. Maybe also add information from a more generic review paper, for e.g.**
https://onlinelibrary.wiley.com/doi/abs/10.1002/9781118682555.ch8

- The reference is added

**P3C1. Three out of four of these are gases, unless you are referring to the particles, in which case, please re-write.**

- The statement is clarified, references are added.

**P3C2. Please provide references that testify that this period was a transitional one.**

- The statement is modified.

**P4C1. Was this performed with AOD estimates or AI estimates?**

- AOD estimates, as mentioned.

**P4C2. What did Su et al. find? if you do not add this information, then remove the reference, there is no point.**

- The main findings are summarized.

**P4C3. What is this acronym? add.**

- Added.

**P4C4. What did they show? which typical regions? you have to actually explain what the articles you reference are all about, not just mention them.**

- The main findings are listed.

**P4C5. As above, what did these studies show?**

- The main findings are summarized

**P4C6. Re-phrase this word. I think "here-after" is more modern.**

- Re-phrased as suggested.

**P5C1. Correct this reference to point to the appropriate ACP[D] paper.**

- The reference is corrected.

**P5C2. Please expand acronym here, since this is the first time you mention it.**

- The acronym is expanded here.

**P5C3. Either Section or Sect. [see line below] please correct according to ACP wishes.**

- Corrected

**P5C4. What does this mean? either expand a bit here or do not mention it at all.**

- Short clarification added; more in Sect.4

**P5C5.**

- Removed, as suggested

**P5C6.**

- Removed, as suggested

**P5C7. Update all such wordings.**

- Updated as suggested.

**P6C1. Move this explanation above, the first time you mention ADV [not in the title]**

- The explanation is moved to Introduction.

**P6C2. I fully understand that this is Part II of a two-part paper series, however some information is required so that the reader interested in this paper is not required to have Part I at hand as well. Two sentences should suffice and is not too much work to add.**

- As you mentioned, Part 2 is a paper of a two-part paper series. Those papers are closely connected and the authors try to convince the reader to read the Part 1 first. Part 1 is already published in ACP. The main findings from Part 1, which are used in Part 2, are discussed in Part 2 in more detail. The ATSR temporal and spatial coverage is not a main conclusion from Part 1 which is further investigated in Part 2. Thus, we do not think that the more detailed discussion on the coverage is needed in Part 2.

**P6C3. Are you sure this is statistically correct? should you grab all the daily values and average them into a season? in this way, all possible ~90 days of the season will have equal weight to the average. In your case, you are basically weighing the three monthly values, one against the other. Also, how were these monthly means calculated? was there a threshold imposed on the amount of daily values required to compute it?**

- We grab all daily values and average them into a month. Seasonal value is the AOD averaged over 3 months. With that, each month has the same weight. We assume this is statistically more correct than to average 90 days of data into the season.

**P6C3. Are these daily L3 datasets? monthly? please add this quite important information here.**

- The important information is added.

**P7C1. Again, I do not think that one should have to go to a different paper to find this rather important information. The location and choice of AERONET sites is important, as important as which AERONET data were used, level 1, level1b, level2? please add a sentence here on which AERONET sites this refers to.**

- Here we just summarize the validation results. The location of the AERONET stations are marked now in Fig.2

**P7C2. Please be more precise, what does close in time mean, 5 minutes or 1 h, for e.g.?**

- Time is specified.

**P7C3. If this main aim of this paragraph was to show the interconsistency of the two satellite datasets, I am sorry to say that you have not reached this goal. Re-phrase this paragraph and re-write adding more information, maybe you can refer to Weatherhead et al., 2017, https://doi.org/10.5194/acp-17-15069-201, as well.**

- Part 1 was aimed to show the interconsistency of the two satellite datasets. The main conclusions from Part 1 are summarized in Part 2. We added the reference to Weatherhead et al., 2017

**P8C1. Re-write this phrase, does not make sense.**

- The phrase is re-written

**P8C2. Please find a more appropriate title for this section.**

- We think that the title fits well to the section, were we discuss the study area and explain the choice for the areas to be compared. We modified the map and moved the section up.

**P8C3. Due to what? land cover? land use? please expand.**

- The discussion is expanded

**P8C4. Re-phrase.**

- The sentence is re-phrased.

**P9C1. The purple lines separating the regions do not show. Also, the figure in general is not very clear. I suggest you increase the resolution.**

- The figure is modified.

**P9C2. There is a question that arises that is not explained satisfactorily: how can one be sure that these are real trend differences and not inter-sensor algorithm differences that affect the trend? the bias between the two datasets, has it been examined? de-seasonalised? is it flat across the years or does it change? a longer discussion is needed here.**

- The bias has been examined in Part I. Deseasonalisation was not needed, as explained in the answer to P2C5. The section is party re-written for clarification and better understanding.

**P10C1. Main questions on the methodology:**

1. **What do you do when the AOD values are too small, for e.g. 0.0001 and 0.0002, which results in a big, unreasonable, and unphysical corrective factor? do you have a threshold that you apply? explain.**

- The numbers you mentioned are too low and never considered. AOD accuracy, according to GCOS, should be within ±0.02. To avoid too high corrective factor, relative correction is introduced (eg. 4 and eg. 5) where the difference in AOD between ATSR and MODIS is scaled by the AOD.

2. **Why are you not weighing the AODs with their respective error estimates? or at least, the std?**

- We follow Sofieva et al., 2017 who makes the next statement: "different amounts of data available over time result in varying uncertainties over time, which might improperly weight the time series. The reference is added

**P11C1. Surely you can spend 5 lines here explaining again. Please recall that Part I is not a published paper, hence one cannot simply reference such a document as if it were the absolute truth. You have to discuss these differences, in short, here.**

- Part 1 is published. The sentence is modified by adding the main reasons for disagreement

**P12C1. The titles within the graph, DJF, JJA, etc, are hard to read.**

- Background for the seasonal titles within the graph is modified

**P12C2. In this section, you definitely have to discuss the expected seasonal behaviour of AOD over China as well as the relative factors that affect it, are the dominant sources biogenic or anthropogenic, dessert dust, etc.**

- This discussion is the main topic of Part I

**P12C3. So, you have a clear steady mean bias of 0.1 on annual averages of between 0.4-0.5? i.e. a 20-25% bias?**
**I suggest you add a figure of monthly mean values before this figure and help discuss the differences of the two satellite datasets otherwise I am not convinces they can simply be "added as one" observational set.**

- Seasonal maps for ATSR and MODIS AOD, as well as maps for difference between two instruments are shown and discussed in Part I. The results from comparison of the validation results is, in our opinion, a prove for the method suggested.

**I suggest you take a long look at the reference list of Weatherhead et al., 2017, and draw knowledge from other similar works, for e.g. https://www.atmos-chem-phys.net/14/6983/2014/**

- Thank you for providing the reference. Even though not referred to that particular manuscript, similar methods have been used in Part I to discuss the similarity in AOD from ATSR and MODIS

**P13C1. You need to provide a reference for this statement and validation results.**

- This statement is made in current paper as a conclusion from the results obtained by the authors.

**P13C2. Re-phrase.**

- The paragraph is re-written

**P14C1. This is not shown in either Figs 5 and 6. Re-phrase or add Figures/statistics to support your statements.**

- Reference to Part 1, supplement, is added

**P14C2. This is not a scientific term, re-phrase accordingly.**

- The whole sentence is deleted.

**P15C1. This entire section feels completely out of balance with the rest of the paper and the aims of the rest of the paper. I suggest to the authors to remove it and keep it for another, more policy-oriented article they might write in the future.**

- The changes in emission policy are only mentioned and not discussed from the political point of view. This short paragraph is needed, since the goal of the manuscript is to show that changes in the emission policy are seen in the AOD measurements.

**P16C1. Are you using the word tendencies because you are analysing P2 which spans only 6 years and you do not wish to say trend?**

- Exactly. Both periods are shorter than discussed in Weathrehead et al.,1988

**P16C2. From this section onwards, I strongly recommend that the authors remove all results/plots/statistics/discussion on the WP. It cannot be accepted as viable statistically.**

- The WP results are removed. The reason to show the WP was to compare the results with the other authors, who consider the longer period of 1-2 decades.

**P16C3. It is impossible to see the green dots.**

- The font for green dots was enlarged.

**P16C4. This range, -0.1 to 0.1, is the magnitude of the bias between the sensors. How do the authors explain this and how can it be shown that this tendency is a real feature?**

- The tendency is calculated for the combined dataset, where the bias between instruments is considered and corrected. The correction depends on AOD.

**P16C5. How do you show results over the sea when you didn't use data over the sea, as you state in the relevant sections? please check.**

- The results over sea is removed. The main reason to not discuss AOD over ocean was the luck of the validation points

**P22C1. I strongly suggest you move this figure, and the associated discussion, before you discuss Fig. 7.**

- In Fig. 7 we introduce the tendencies over whole China pixels-wise (L3) and show where the tendencies are significant. Then we compare AOD tendencies for selected regions. Our logic is to go for details (comparison between regions) after introducing the whole picture.

**P24C1. This graph should also move further in the front of the paper. It is also rather too busy with all the values/statistics/asterisks etc.**

- Please, see our reply to P22C1.

  We agree that the graph is busy but we consider that all information shown is complementary for better overview and the graph is readable. It will take some effort from the reader to read the corresponded numbers in the table.

**P26C1.**

- The statement is deleted.

---

## Author Comment (AC2) · 6 Sep 2018

We thank the reviewer Dr. A. M. Sayer for his positive statement and very constructive comments. We much appreciate his thoughts and suggestions that reflected in the improvement of this manuscript. Detailed answers are below.

**Summary:**

I am writing this review under my own name (Andrew Sayer) as I have previously discussed this research with the authors, and am on the team responsible for the MODIS aerosol data products being used in the study. I also reviewed the paper de Leeuw et al (2018), which is in some sense a predecessor to this study, and the Part I of this paper also by Sogacheva et al and also currently in ACPD. I feel I am able to provide an impartial review, but am signing the review in the interests of transparency.

The goal of this pair of papers is to look at spatial and temporal (seasonal/interannual) variations of AOD over China. This is accomplished mainly by using two satellite data sets: the ADV algorithm applied to the combined ATSR2/AATSR record (1995-2012), and the combined Deep Blue/Dark Target algorithms applied to the MODIS Terra record (2000 onwards) from the latest Collection 6.1. Part I contains some validation results and an initial look at the time series, while this Part II focuses on trends ("tendencies" in the authors' terminology) during several time periods where emissions policies may have influenced the aerosol loading. These papers are linked so I will summarize my review of Part I first (which be found on the ACPD page at https://www.atmos-chem-phys-discuss.net/acp-2018-287/ ), since this Part II requires Part I to stand on. For Part I, I have recommended revisions and re-review. The two main technical threads of my review of Part I were that (1) more needed to be done to establish the validity of treating ATSR2 and AATSR as a single record (which is the underlying but untested assumption), and (2) some of the time series analysis in Part I should be moved to Part II to keep the flow of both papers better and avoid some redundancy. So this review should be read with that in mind.

My overall recommendation for this Part II is also for major revisions and re-review. It's an interesting and important topic, but I don't think it is ready for publication in current form. I would like to review the revision; I would prefer if Part I can be revised and eventually accepted for publication first, if possible, so that we have that as a stable version to refer back to when reviewing a revision of this Part II, since the papers are quite closely linked. This is an interesting study but I think (see below) that the ATSR/MODIS merging technique requires some more examination, and also the conclusions would be better supported by including additional meteorological and/or geophysical data products in the analysis (so we can see whether AOD changes are likely to be the result of policy, or whether weather patterns may be an influence here).

Uncertainties in the method and results also need better quantification. Note I am not an expert on policy or emissions, so my comments mostly focus on the statistics and AOD data. Hopefully another reviewer can comment on policy/emissions in more detail – my lack of comments is due to a lack of expertise to judge in those areas.

The quality of language is overall good and any issues can probably be dealt with by Copernicus' copy-editing and typesetting process. Therefore my review mainly concentrates on technical abstracts. I have tried to separate each main comment into its own paragraph to respond to. Here, PXLY refers to page X, line Y.

Specific comments:

Abstract: I would condense this into one paragraph if possible and shorten it to highlight the main findings. For example I think the authors can cut out the discussion on linear trends across the whole period, since one of the main points of the study is that it should not be considered one period due to the changes in emissions policy. I'd also cut out the discussion of annual trends/tendencies since I think (as discussed before, and below) only seasonal trends are meaningful due to the seasonal differences in aerosol loading, type, and retrieval coverage (from e.g. cloud cover, snow) aliasing into the annual means in a complicated way. Also, the papers cited in the abstract can be removed – these citations are in the body text, they are just adding length here; traditionally one doesn't need to provide citations to back up statements in the abstract because that's what the rest of the paper is for.

The Abstract is shortened. The citation is removed.

P3L2-3: I would avoid giving urls like this as citations here, particularly since the latter is an opinion piece. Urls are not always stable and one can't be sure the content is going to change or is valid. It would be better to cite something with a DOI or official publication number. For example the first link is for the World Bank so there must be some report or something which can be used.

URL is replaced with a reference to the similar publication.

Figure 1: Likewise, I would not give an url here for the population data used. If you click through the url, it gives a citation for the data set which should be used instead.

The DOI for the data is provided.

A couple of other things jump out at me from this figure. First, it seems that the largest population change in this region is not in fact China, but India. If population acts as a driver for anthropogenic aerosol emissions, one might expect that observed aerosol changes in China may be influenced by changes in transported aerosols from India. If this contribution cannot be quantified, it means that one cannot state that observed changes in China are a result of changes in Chinese policy. (The fact that aerosols don't follow

**national borders is one reason why in general I prefer regional studies to national studies – you have to be able to account for the broader context of regional emissions/meteorology changes.)**

We agree with the hypothesis that aerosol measured in China might be transported from some other areas (India, Russia, etc.). However, on averaged AOD maps, AOD level over SW China is low, since the AOD transport from India and Bangladesh, which are the strong source of aerosol particles of different origin, is highly blocked in eastern direction by the Himalayas. Whereas highly populated and industrial areas are often recognized in China by the local elevated aerosol concentration. The increase in the population usually follows the growth of industry and, in case of China, is resulted in the increase of pollutants. In the current ms, we consider different areas, where the economic growth was not equal during the last decades. Difference between the areas in the aerosol tendencies proves that changes in local emissions play stronger role than the change in the aerosol transportation to China from outside the country. We mention later in the text other reasons for changes in the aerosol concentration, but the impacts of each factor are not considered.

**Secondly, it looks like the population in the Sichuan Basin area (30 N, 105 E) has dropped somewhat since 2000, while the rest of China has been flat or steadily increasing. Is this right? I did a quick search online and it looks like the Chengdu metropolitan area population is increasing (http://worldpopulationreview.com/world-cities/chengdu-population/ - perhaps this is the red dot on the map here – but the overall population of Sichuan province is fairly stable (http://population.city/china/adm/sichuan/ ). However I'm not sure of the reliability of these sources, and it is difficult to estimate the total population from these maps of population density because the colour scales seem to saturate and we don't have grid size information. So perhaps people in Sichuan province are becoming more concentrated in Chengdu, I don't know. The point is, I think both of these aspects should be discussed in more detail in the manuscript.**

You are right that overall the population in the Sichuan province has not changed much. However, people were moving to the Chengdu metropolitan area (current color scale, which was chosen to be the same for all three maps for comparison may not show that clearly) from other regions of the Sichuan province. Thus, we discuss in the text also the changes in population in megacities. We added some clarification to the text.

**Section 2: This is largely a repetition of Part I, in that it is introducing the satellite data used. I understand a recap of the data is needed, but I think that this could be shortened. For example we don't need to know the spectral and spatial resolution of the ATSR and MODIS instruments, or provide the validation summary table. I think it's enough to say you're using the level 3 monthly products at 1 degree, and refer back to Part I for more details. Particularly since the number of 3-way matchups (AERONET, ATSR, and MODIS all together) was low and largely confined to cities in Eastern China, I think discussing these statistics in detail gives a (possibly false) impression that we can be confident that these are representative of relative performance across the whole of China. Plus, instantaneous validation differences will not necessarily reflect differences in the monthly or longer means, and the method in Section 4 is meant to show and reconcile these differences. So I think Section 2 can be shortened to a couple of paragraphs.**

The section is modified. The note about the limited number of validation points in the northwest of China is added. We keep the summary for validation results, on which the method introduced here is based.

**Section 4, general: the large number of long subscripts on variable names makes things cumbersome to read (hard to visually follow the equations). I suggest replacing "ADV" subscripts with "A" and "MODIS" or "MOD" (both are used, but I think mean the same thing) subscripts with "M". Other subscript shortenings could include "y" for "year", "c" for "comb", or "rc" for "rel_corr", for example. This will make it easier to follow the**

**equations.**

Some of the subscripts are shortened, as suggested. Others are kept as originally introduced since with further shortening the visualization makes reading more complicated.

**My understanding of the method is that the overlapping ATSR/MODIS period (2000- 2011) just averages the two time series. Then the size of this adjustment in the overlapping period is used to scale the pre-2000 ATSR data, and the post-2011 MODIS data, to generate the "merged" time series. This is done twice: once for an "annual" correction, used for the merged multiannual time series, and once for a set of four "seasonal" corrections, used for the merged seasonal time series. Further, the calculation is performed separately for each 1 degree grid cell. Is that a correct description? If so I would include a couple of sentences to that effect somewhere up the top, before the equations, in case the reader gets lost.**

The text was modified. In the current version, we first introduce the method and then say that it is applied to L3 annual and seasonal aggregates.

**P10L27-28: the authors mention Bourassa et al (2014) as an example of this technique being used elsewhere. However I'm not sure that is necessarily a good justification. Bourassa et al were looking at stratospheric ozone, which has (to my knowledge) a somewhat longer lifetime and smoother spatial distribution than AOD, as well as fewer contextual (i.e. surface cover or type-dependent) uncertainties. These are very different error characteristics. Looking through the Bourassa paper, the relative differences between the ozone data sets used were in many cases a lot smaller than the differences seen for AOD here. So the method which is justifiable for one geophysical data set is not necessarily a good choice for another one. Are there other example applications, or justifications which can be made?**

We agree that the approach for combining ozone data sets is different (https://www.atmos-chem-phys.net/17/12533/2017/acp-17-12533-2017.html). We removed the reference to Bourassa et al. (2014). We could not find any reference where a similar method to combine data from different satellites is introduced and discussed.

**I have some more general concerns about this method for combining the ATSR and MODIS records, which become more severe because the purpose is trend analysis, which becomes particularly sensitive to values at the start and end points. Both the start and end points of the combined record are being adjusted by this method, and the "combined" to "adjusted MODIS" changeover also takes place at the same time (year 2011) as the start of the trend analysis period "P2" (2011-2017). So uncertainties in the merging process will unfortunately affect the trend analysis at the points at which a trend calculation is most susceptible to artefacts.**

We indeed did not find the way to estimate the uncertainties related to the merging. However, we estimated the difference in the tendencies, assuming that the combined AOD is over- or underestimated by 10%. In the figure below, we show the annual AOD tendencies for P1 (upper plot) and P2 (lower plot) estimated for each region from the combined dataset (black line) and for two "tested" AOD datasets. In the first tested dataset we assume that our combined dataset is overestimated and "real" AOD is 10% lower (blue line). In the second dataset, we assume that our combined dataset is overestimated and "real" AOD is 10% higher (red line). For both P1 and P2, the difference in the AOD tendencies for the tested datasets was less than 10% from the tendency estimated of the combined dataset.

[Figure]

However, we decided not to include those checks into the manuscript, since, as it was mention before, the length of the periods, where we estimate tendencies, is short and the results might have uncertainties due to the short length of the periods (thus, called tendencies).

Instead, to give an estimation of the quality of the combined AOD datasets, in the revised version we added a chapter (Sect. 3.3) on the ADV, MODIS and combined AOD seasonal/yearly comparison with AERONET. This is not validation but more a comparison, since all the AERONET available data were used to calculate the seasonal aggregates and the satellites temporal coverage is lower. It means that some of the events might be missed by the satellites which are included in the AERONET AOD seasonal/annual aggregates. The comparison has been performed for four periods:

- T2 (2000-2011, overlapping)

- T3 (2011-2017)

- ADV period (1998-2012)

- MODIS period (2000-2017)

For T1 (1995-2000) there was not enough AERONET data for comparison.

The comparison shows that the statistics for the combined dataset are slightly better than for MODIS and ADV separately.

**I appreciate the authors' efforts to merge the ATSR and MODIS records here, and I don't know that there is any well- defined most appropriate way to do so. So this is one attempt which seems like a reasonable thing to try, and I am not sure what other way to harmonise the AOD record to suggest. However the fact remains that this method will introduce uncertainties, which may be systematic, and influence calculated trends.**

We calculated the AOD standard errors for each year/season for both ADV and MODIS but, we decided to not include it to the manuscript. Here is an example for annual AOD :

[Figure]

AOD standard error, year

For China and SE China, the AOD standard error is low (~0.01). For other areas, the AOD standard error is higher, but what is important here is that the ADV and MODIS AOD standard error time series have similar patterns overall. This gives us the confidence to conclude that the AOD standard errors do not influence the AOD time series significantly.

**So, at a minimum, the possible influence of these effects must be quantified. Here are some thoughts about how to do that. The decision to do a simple average for the overlapping (2000-2011) period seems to have been made on the fact that the mean bias of the two data sets vs. AERONET, based on the limited available number of samples and limited available locations of samples, was roughly equal and opposite (so averaging might be expected to cancel out the bias on an average basis). Despite the fact that this was presented as a difference in ABSOLUTE AOD bias, the correction term is applied as a RELATIVE AOD correction. So that feels somewhat inconsistent.**

We scaled the correction to avoid the situation, when the correction itself is higher than the AOD.

**The authors could make maps of MODIS-ATSR on a monthly basis, both in absolute and relative terms, and see what it looks like. If the bias vs. AERONET of roughly 0.06 (for MODIS) and -0.07 (for ADV) is representative everywhere, then I would expect these maps of absolute difference to hover around values of 0.13 and have little spatial or temporal variability. If they don't, then it tells you that the AERONET matchups aren't representative of the bigger picture. In contrast if maps of relative AOD show small variability, it suggests that a relative scaling is more appropriate.**

We disagree that the absolute difference of ~0.13 should be representative everywhere. This number depends on the sampling. For fine-dominated aerosols, the validation results show a bias of -0.09 and 0.08 (for ADV and MODIS, respectively); for coarse-dominated aerosols, the bias is -0.11 for ADV and 0.10 for MODIS. The conclusion we draw is that biases for ADV and MODIS are opposite in sign but similar in amplitude. We modified the conclusions in Sect. 3.3.

**For the period with both ATSR and MODIS data (2000-2011), my understanding is that the two time series are averaged, and it is the correction across this 11-year period which is the basis for correction periods T1 and T3. So one other thing which could be tested is to see what the variability in the MODIS-ATSR difference is over 2000- 2011, e.g. what the standard deviation of the difference is. If it is 0 then MODIS and ATSR are always offset the same amount. In reality it will be nonzero, and this additional variability should be propagated into the trend uncertainties discussed later in the paper. Although you might get a trend with apparently low noise, if you know that part of your time series may**

**have uncertainties which aren't captured by this error model, then those errors (in this case, the interannual variability in MODIS-ATSR AOD in 2001-2011) should be added on when estimating the total uncertainty on a trend. A similar point can be made (see my review of Part I) when considering the combination of ATSR2 and AATSR to give the single combined ATSR record used as the basis here.**

We estimated the standard error for MODIS-ATSR difference over 2000-2011. See the figure below.

[Figure]

For the annual AOD, the dAOD standard error is around 0.01; for the seasonal AOD, the error is somewhat more significant (ca. 0.02 with some deviation for different seasons and regions). We consider that it is low value and will not change the tendencies significantly (as shown above, where the difference between AOD from the combined data set and the tendencies for the combined ± 10%AOD are presented).

In Sofieva et al., 2017 the uncertainties are not considered (https://www.atmos-chem-phys.net/17/12533/2017/acp-17-12533-2017.html), where trends in ozone are estimated. They make a statement that "different amounts of data available over time result in varying uncertainties over time, which might improperly weight the time series. In our regression, all data points are considered with equal weights, and the uncertainty of the fitted parameters is estimated from the regression residuals". We are following Sofieva et al. method, and also added uncertainties for the estimated tendencies to the table with the statistics for tendencies.

**My remaining comments are more general, because I think that the above comments and discussion, plus the review of Part I, may necessitate a rewrite of some of the later sections of this study. Some of the points which I think need to be discussed here in more detail include:**

**1. Trend terminology. The authors say "tendency" rather than "trend" throughout. I prefer the more standard "trend". From previous discussions with the authors, my understanding is that they preferred the term "tendency" because they feel that "trend" should refer to a longer time period than considered here. My personal feeling is that "trend" is clearer for the reader, so long as it is made explicit that one should not extrapolate out of the time period under consideration.**

Indeed, we say "tendency" rather than "trend" because the length of the periods when AOD is more less steadily changing is short (11 and 6 years) to estimate trends (Weatherhead et al., 1998).

**2. Annual and whole period trends. I continue to think that, since the time series show seasonal variation and are not linear, it is not sensible to do whole period (WP) trends, or annual trends. I think it's best to show only the piecewise trends.**

The results and discussion on the WP AOD tendencies were removed. We included those into the version submitted to ACPD thinking that they could have been used for the comparison with other studies (e.g., Wang et al., 2017). However, we keep the results and discussion on the annual AOD tendencies, since this general information is useful for e.g., modelers.

**3.  Trend breakpoints. The authors split the trend analysis into WP, an early P1, and late P2. These split times are informed by times where policy changes may have had an influence. However there also exist established statistical methods to estimate  whether there are breakpoints in trends, and when these breakpoints are. It would be  good to use these methods to see whether in fact any such breakpoints are detectable  at the point the authors assume they are there.**

In the current manuscript, we study the possible connection between the emission policy and the aerosol concentration. Other factors, e.g., meteorological conditions, dust transport, biomass burning are discussed in Part 1 and Part 2 briefly.

Besides binding the choice for the periods to the Five-Years Plans and emission reduction policy, we performed statistical tests, where we looked at the AOD tendencies, uncertainties of the tendencies, the error of the slopes and tendencies relative error annually and for all seasons all regions. The example for the statistics for annual tendencies for periods of different length starting in 1995 is shown below (number at, e.g., 1999 show statistics for the period 1995-1999).

[Figure]

Similar test was performed for the second breakpoint in ~2011. We added a paragraph to the manuscript (Sect. 5.3), summarizing the results.

**4.   Trend significance. The actual fitting mechanism and uncertainty estimation is not discussed in detail. P21L5 says it is linear regression (I assume ordinary least squares) and they define p<0.05 as statistically significant (I assume here this is de- fined as the estimated trend is at least twice as large as the estimated uncertainty in that trend, i.e. 2 sigma). Is autocorrelation considered? Often in trend studies lag 1 autocorrelation is estimated and used to correct the uncertainty estimate (because many  geophysical data fields can be autocorrelated on time scales of months, seasons, or years). This gives a more realistic, and generally larger, estimate of the uncertainty  on the trend because**

**autocorrelation tends to be positive. See e.g. Weatherhead et al (JGR 1998, https://agupubs.onlinelibrary.wiley.com/doi/pdf/10.1029/98JD00995 ). If this is not done then there should be some evidence given why this model is reasonable.**

We have not include autocorrelation into our tendency analysis. The reason for that was that we do not expect any autocorrelation between yearly AOD (as, an example, one exists for other variables dependent on, e.g., 11-years solar cycle). Our annual time series are built from annual values only. Seasonal time series are built from correspondent seasonal aggregates, thus there is a one-year lag between neighboring values.

**Further, as noted above, the uncertainty from the ATSR/MODIS merging exercise should ideally be considered.**

As mention before, we did not have a possibility to estimate the uncertainties. The results for seasonal/interannual comparison with AERONET, which are included in the revised version, gives a rough estimation on the combined AOD quality.

**Additionally, there's the problem that since so many comparisons are being performed there are likely to be a number of false positives (trends which appear significant but are coincidental). See e.g. Wilks (JAMC 2006, https://journals.ametsoc.org/doi/10.1175/JAM2404.1 ) for more on the false discovery rate and how to deal with this sort of thing.**

Thank you for the suggestion. In the current manuscript, the AOD *tendencies* were discussed; we apply the statistical significance test for both L3 AOD time series fitting and the averaged over the selected areas AOD. We will consider the method introduced in Wilks when the length of the period will be long enough to discuss *trends.*

**5. Presentation of trend uncertainties. In the later Figures and discussion, trends are given in terms of both absolute and relative AOD (where I guess relative AOD is defined with respect to some base year – this is unclear – since as the AOD changes by some absolute amount per year, the relative trend would change while the absolute trend would not). For clarity and comparability between regions, I would rather just see absolute AOD trends.**

We think that the maps with the AOD relative tendencies contain information that is complementary to the AOD absolute tendencies. With those, we can discriminate the areas where the AOD had the higher changes compared with the AOD local values. This information can be potentially compared with relative changes in the emissions (currently we do not have access to that information on the emissions).

**I would also like to have the uncertainties estimated on the trends be presented and discussed (the Appendix tables say they have absolute error in percent, but it's not clear to me what exactly that means, and it is more relatable to have in absolute units like the trend itself). For example at face value the sign of a trend may have changed between two time periods, but it may be that the difference is within the uncertainties of the individual trends, in which case it is more accurate to say that a change in behaviour cannot be identified. Thresholds on p-value are somewhat arbitrary and so I think it is generally more useful to present trend uncertainties instead (or in addition) in the text and tables in the Appendix.**

We added the results for tendencies uncertainties to the Table A1 and discussed them in the text.

**6. Evidence of attribution for trends. It makes sense that changes in policy could affect emissions and change the AOD. However another key factor, which is examined in several other studies in China and elsewhere, is changes in meteorology. For example if there have been changes in air stagnation frequency, or aerosol transport pathways (for sources outside China as well as those inside China), then these might be magnifying or masking any trends resulting from policy changes. Additional data sets that might be able to support this would include emissions data bases, other satellite products (e.g. SO2 and NOx, which are briefly discussed), and meteorological reanalyses. Some of this has been**

**done by other studies, and I'd like the discussion to go into more detail on those. But some of this may not have been and the authors might need to do these analyses themselves. Otherwise it is premature to try to state the reason for the trends.**

In de Leeuw et al. (2018) and Part 1, we briefly discussed the potential impact of the meteorological conditions and the dust transport on the AOD in China. In the introduction to Part 2, we also shortly mention the results from other studies on the contribution of the meteorological conditions on the AOD in China. E.g., Gu et al. (2018) are referred, who shows that the variation of AOD over Beijing was significantly affected by the anthropogenic aerosol emissions and less affected by the wind and temperature inversion. In the current manuscript, we show that the AOD tendencies have a good agreement with the 5-years plan related emission policies. We discuss the changes in the emission policy in China and briefly discuss the changes in NOx and SO2 emissions in China published by van der A (2017). The detailed analysis on the decoupling of the contribution from changes in the emissions and the meteorology, including dust transport, and their influence on NOx, SO2, and AOD over the whole China, which implies a detailed analysis of the reanalysis data, is a topic for another manuscript.

**In summary, the main thread is I feel it is important to be thorough and give reasonable uncertainties – say what we can – than to make a conclusion which isn't fully supported. Particularly since this is an inherently political topic. Maybe the data we have are not enough to be conclusive yet, in which case it is even more important to say as much as we can but no more and be clear about what the biggest uncertainties which we need to reduce to be able to answer the question are. The last sentences of the paper (P26L26-27: "Thus, in the current study the effect of the changes in the emission regulations policy in China is evident in AOD decrease after 2011. The effect is more visible in the highly populated and industrialized regions in SE China.") are very strong statements, and this might indeed be the case. But I don't think the discussion of uncertainties is quite thorough enough, or other explanations and their contributions examined in enough detail, to make this case.**

We agree that the statement is strong, but we are not claiming that the AOD decrease is caused by the changes in emission policy only (we discuss other factors which may influence AOD in the introduction). "Evident", in the current sentence, means that the AOD decrease is noticeable after 2011, when a new regulation plan for emissions was accepted. Those two events (new police and the decrease of the AOD) are in good agreement. The contribution of the anthropogenic emissions to the AOD is critical in China (however, might be comparable with episodic dust or biomass burning transport). Such a strong decrease cannot be caused by e.g., meteorological factors only, which contribute less to the AOD, as compared to the anthropogenic emission in China, as shown by Gu et al. (2018).

---

## Author Response (AR2)

**To the Reviewer Andrew Sayer:**

We would like to thank Andrew Sayer for his criticisms and suggestions to the manuscript (marked in *Italic* below). Most of the modifications suggested were considered in the revised version of the manuscript and below are the comments to his questions/suggestions.

*I (Andrew Sayer) reviewed the previous iteration of this manuscript, as well as the related Part I which was recently published in ACP. The authors have put a lot of effort into their revisions of both Part I and Part II which I feel improve the flow and content. This is an interesting and informative analysis and I think the authors have done a lot of good work; my comments below reflect the fact I want to help make the paper the best it can be, and there are a few points where the current text still has issues. I recommend another round of revisions and if the Editor would like would be happy to review the next version. There are three main areas where I think further changes are necessary:*

*1. Generally I think that the authors' statements on policy attributions for changes in AOD still come across as too strong, since they aren't directly examining the related emissions/meteorology patterns directly here (though they do refer to other studies for some of that information). From the authors' response it seems that making strong attributive statements was not their intention so I wonder if some of this comes down to language differences on how we interpret the intensity of certain words. But for me, for example, the final paragraph of the abstract is a very strong attributive statement: "The long-term AOD variations presented here show that the changes in the emission regulations policy in China during 1995-2017 result in a gradual decrease of the AOD after 2011 with an average reduction of 30%-50% between 2011 and 2017. The effect is more visible in the highly populated and industrialized regions in SE China, as expected." Similarly, the final paragraph of the paper: "Thus, in the current study the effect of the changes in the emission regulations policy in China is evident in the AOD decrease after 2011. The effect is more visible in the highly populated and industrialized regions in SE China." I would suggest deleting these sentences entirely or at least replacing strong statements like "the variations ... show that the changes in the emissions regulations policy ... result in... " and "the effect of the changes in the emission regulations policy in China is evident" as (for me as a reader) those words are making a direct explicit link, which could be true, but is not supported by the analysis in the study. It would absolutely need additional analysis including emissions estimates, satellite/ground data on trace gas precursors for aerosols like SO2/NOx, meteorology etc. Yes I know that other studies have looked at these in isolation but if you want to make a statement about attribution you need to show it all together. That could be a follow-up study. I don't see why there is a need to make a strong statement about attribution in this study anyway?*

We agree that the statement about the direct relation of the AOD reduction to the emission policy regulation is too strong. The statement was removed from the abstract and conclusion.

*2. One thread of my previous review was to better understand the uncertainties in the trend calculation resulting from combining the time series from the two (quite different) retrievals. The authors have not been able to quantify the additional uncertainty which this adds, noting (correctly) that this is not a simple task to do so. I also suggested they adopt the methodology of Wilks to assess/remove false positives in the significance calculations due to multiply hypothesis testing, which they chose not to do. In light of both of the above, I think that discussion about statistical significance (here I think it is based on trend/tendency magnitude being more than twice the estimated uncertainty on it, which is saying that the chance of this happening if there is no trend is 5% or less) should be removed from the paper. The p<0.05 approach as an arbitrary threshold for whether something is statistically significant is so well-used that it has become a gatekeeper for whether or not a given result is talked about (and other research communities such as psychology/medicine are currently undergoing something of a paradigm change in terms of dealing with this), and in this case we know (and the authors have stated in their response) that there are components of the trend uncertainty that they do not include because they cannot model them well. In that case we know that the total uncertainties are likely to be underestimated by unknown magnitude, and as a result estimates of the statistical significance of the result are overestimated by an unknown magnitude. Because of this I think that it is better to present not trend estimates and their significance but rather trend estimates and their best estimate of the uncertainty, and to acknowledge that this uncertainty is a lower bound due to these issues. This would obviously require changes to the manuscript throughout (to text, figures, and tables; maybe for some figures panels showing uncertainty could be added, or filters for maps could be based on absolute uncertainty in the trend rather than an estimated significance level), but seems to be a fairer presentation which avoids a reader potentially interpreting a result as stronger (as in, more likely real than due to chance) than it is. I do not see an advantage in including the significance values when we know that they are biased estimates of significance and know that they may end up misleading a reader who is not statistically-minded or does not go through thoroughly.*

We agree with the Reviewer that the p<0.05 approach might be misleading in the case of the short length of the time series. However, the p-approach results were included to the discussion paper since the results looked reasonable for us (AOD tendencies were confident in the areas with the good coverage and high variability of the AOD). As suggested by the Reviewer, the results for the confidence of the tendencies are now removed from all figures and tables, as well as from the discussion.

*3. As the authors know I am strongly against least-squares linear regression for AOD validation scatter plots, because although it is a common technique it is statistically invalid here, as the data violate most/all of the requirements for the technique to be applicable (see e.g. http://people.duke.edu/~rnau/regintro.htm or statistics textbooks). The slope/intercept coefficents and estimated uncertainties you get up are known to be wrong and systematically (not randomly) biased in such cases. I don't see any scientific justification for including something we know to be wrong in the paper. I also don't think they are a useful interpretive aid for e.g. Figure 5, since the data volume is fairly low, and the binned analysis (circles with error bars) here convey related information (i.e.,*

*magnitude and sign of the bias in low-AOD and high-AOD conditions) in a much better way. Please just delete the regression slopes/intercepts and discussion from the paper.*

The linear regression slopes and intercepts, as well as related discussion, are removed, as suggested.

*I also have a number of more minor issues*

*P2L21: I don't think that "total suspended particle matter (TSP)" is needed here – it would be better just to write "particulate matter" and the grammar isn't quite right written this way ("Coal smoke mainly contains total suspended particle matter" doesn't*
10 *make sense while "Coal smoke mainly contains particulate matter" does).*

The sentence was corrected, as suggested.

*Figure 2: based on the text, the labels for the Taklamakan Desert and Tibetan Plateau (8 and 9) are inconsistent. (I think this was*
15 *also the case in the original submission but I did not spot the error then.)*

The labels on the map are correct. We corrected the wrong names on p.5, lines 18-19.

*P8L14: I think that there are some missing numbers about fine-dominated error here (looks like only the coarse condition results*
20 *are shown), based on the later text on P9 L30-31. These two sections are also repetitive; I'd just delete this sentence on P8L14 as I think the information fits better on P9.*

On P9, we give an overview for the ADV and MODIS validation results. In Sect. 4.3 we show the results for the combined AOD seasonal/annual aggregates comparison with AERONET. In the revised version, we clarify that in the section title and in the text.
25 Since the number of points is limited and to avoid additional uncertainty from estimating the aerosol types for seasonal aggregates, the AOD types are not considered in the current exercise.

*Figures 7, 8 (as well as general comment on the text throughout): in addition to my significance comment above, these two figures are somewhat repetitive in that it's the same thing just expressed on an absolute scale in Figure 7 and a relative scale in Figure 8.*
30 *I am not sure that Figure 8 adds much, and suggest deleting it. A small (and scientifically unimportant) change over a fairly clean background can appear prominent as a small change to a small number can appear big, while a more modest increase/decrease over a polluted area (which might represent a real policy success or failure) in contrast fades from prominence as it becomes a smaller relative change. In other words, getting 100% richer is quite different in real terms if you only had $1 to start with compared with if you were a billionaire to start with. Basically I'm saying that Figure 8 could distract from where the interesting results are*

*happening. In general I am in favour of absolute rather than relative AOD changes throughout for that reason (and I think the other reviewer had expressed comments against writing things in relative terms). Removing the stuff about relative changes would also make text and figures more readable by making the sentences more concise and removing visual clutter. I don't see an advantage as a reader to including both.*

We understand the reviewer's concerns about the wrong impression on high relative tendencies, which might appear in the case of low AOD. However, we consider the information about the AOD relative change useful for better understanding the AOD dynamics. To avoid a high but scientifically unimportant AOD relative change over a fairly clean background, Figure 8 was replaced with the other one, where the results are shown if the mean AOD was above 0.1 for the corresponding period.

**To the anonymous Reviewer:**

*Dear Authors,*

*Congratulations on performing such a comprehensive revision based on the two referee comments and suggestions. The manuscript*
15 *is much improved now, however my main concern on the statistical/numerical results remains unaltered. A very high negative tendency (for e.g. of -9%/annum or -90%/decade) can easily be misappropriated by other works, who will not go into the details of how these numerics were calculated [i.e. that they are based on six numbers, for the short period.] Maybe another careful consideration on the summary of the statistical findings might be performed.*

20  We would like to thank the Reviewer for his positive comments on the revised version of the manuscript.
To avoid the misunderstanding of the results, in the last version of the manuscript we clarified another time in the text that the statistics are reported  for two periods, 1995-2006 and 2011-2017.

[revised manuscript text omitted]